# Blockade of the LRP16-PKR-NF-κB signaling axis sensitizes colorectal carcinoma cells to DNA-damaging cytotoxic therapy

Xiaolei Li[1†], Zhiqiang Wu[1†], Xiaojing An[2,3†], Qian Mei[1], Miaomiao Bai[1], Leena Hanski[4], Xiang Li[1], Tero Ahola[5], Weidong Han[1*]

[1]Department of Molecular Biology, Immunological and Bio-therapeutic, Institute of Basic Medicine, Chinese PLA General Hospital, Beijing, China; [2]Department of Pathology, Chinese PLA General Hospital, Beijing, China; [3]Department of Pathology, Xiyuan Hospital, China Academy of Chinese Medical Sciences, Beijing, China; [4]Division of Pharmaceutical Biosciences, Faculty of Pharmacy, University of Helsinki, Helsinki, Finland; [5]Department of Food and Environmental Sciences, University of Helsinki, Helsinki, Finland

**\*For correspondence:**
hanwdrsw69@yahoo.com

[†]These authors contributed equally to this work

**Competing interests:** The authors declare that no competing interests exist.

**Abstract** Acquired therapeutic resistance by tumors is a substantial impediment to reducing the morbidity and mortality that are attributable to human malignancies. The mechanisms responsible for the dramatic shift between chemosensitivity and chemoresistance in colorectal carcinoma have not been defined. Here, we report that LRP16 selectively interacts and activates double-stranded RNA-dependent kinase (PKR), and also acts as scaffolds to assist the formation of a ternary complex of PKR and IKKβ, prolonging the polymers of ADP-ribose (PAR)-dependent nuclear factor kappa B (NF-κB) transactivation caused by DNA-damaging agents and confers acquired chemoresistance. We also identified a small molecule, MRS2578, which strikingly abrogated the binding of LRP16 to PKR and IKKβ, converting LRP16 into a death molecule and forestalling colon tumorigenesis. Inclusion of MRS2578 with etoposide, versus each drug alone, exhibited synergistic antitumor cytotoxicity in xenografts. Our combinatorial approach introduces a strategy to enhance the efficacy of genotoxicity therapies for the treatment of tumors.

## Introduction

Death in patients afflicted with cancer is mainly due to uncontrollable recurrence of the primary tumor following failure of current therapies. Colorectal cancer (CRC) is one such cancer type wherein aggressive treatment strategies including surgery, ionizing radiation (IR), and chemotherapy provide only palliation (*Benson et al., 2004*; *de Gramont et al., 2000*). The most important factor limiting the success of systemic anticancer therapy in achieving cure or prolonged overall survival has been drug resistance, either because the initial tumor fails to respond to therapy or because it acquires resistance during relapse (*Holohan et al., 2013*; *Kuczynski et al., 2013*). The anticancer activity of most chemotherapy drugs relies on the induction of DNA damage in rapidly cycling tumor cells with inadequate DNA repair. However, damage induced by drugs is not invariably lethal but instead actively triggers damage responses, and it is these responses that determine the eventual fate of the cell (*Johnstone et al., 2002*). Given the adaptability of tumor cells, it seems likely that drug resistance will continue to be an important clinical problem, even in the age of targeted therapeutics and tailored treatment regimes (*Lowe et al., 2004*). Thus, novel therapeutic strategies that circumvent the mechanisms of resistance are urgently required to improve the survivorship of cancer patients.

**eLife digest** Most chemotherapy drugs kill cancer cells by damaging their DNA. The cells have systems to combat this damage and help them to survive, and in some cells these systems work effectively enough to make the cancer effectively resistant to the treatment. For example, a protein called NF-κB can turn on various genes that help to repair damaged DNA. However, DNA is contained the cell nucleus, whereas the inactive form of NF-κB is found outside the cell nucleus. So how does the damaged DNA communicate with – and activate – NF-κB?

Previous research had found that another protein called LRP16, which resides in the cell nucleus, plays a crucial role in the repair process that NF-κB is involved in. Li, Wu, An et al. have now studied bowel cancer cells taken from human tissue samples and found that the cancerous cells contained higher levels of LRP16 than cells from the surrounding tissue. Patients with cancers containing very high levels of LRP16 were more severely affected by cancer. Further investigation revealed that when DNA is damaged, LRP16 moves out of the cell nucleus and stabilises how NF-κB interacts with two other proteins; this stabilisation activates NF-κB. LRP16 therefore appears to regulate the signal that travels out of the nucleus from the damaged DNA to activate NF-κB.

Further experiments showed that anti-cancer treatments worked best on cancer cells that lacked LRP16. Thus it appears that LRP16 helps cancer cells to respond to and resist the DNA damage caused by chemotherapy. Li, Wu, An et al. went on to identify a drug that prevented the activation of NF-κB by blocking the effects of LRP16. Using this drug alongside chemotherapy drugs made the cells more likely to self-destruct. More work is now needed to develop therapies based on the newly identified drug and to establish how DNA damage activates LRP16.

Our understanding of the mechanisms that protect tumor cells from the cytotoxicity induced by drugs must be improved.

The nuclear factor kappa B (NF-κB) signaling pathway, which is activated by many stimuli, is a crucial transcription factor in a variety of pathophysiological conditions (*Harhaj and Dixit, 2012*; *Hayden and Ghosh, 2008*; *Smale, 2011*; *Wan and Lenardo, 2010*). NF-κB responds to genotoxic threats via the activation of the inhibitor of NF-κB kinase (IKK) and NF-κB liberation from IκB proteins, similar to the canonical pathway activated by external stimuli (*Janssens et al., 2005*; *Perkins, 2007*; *Wu et al., 2006*). The NF-κB signaling pathway has emerged as an important mediator for cellular responses to DNA damage, in particular NF-κB-conferred anti-apoptotic transcription facilitates the escape of cells from the lethal effects of DNA damage (*Janssens et al., 2005*; *Perkins, 2007*; *Wu et al., 2006*), and initiates cell cycle checkpoint control to promote cellular recovery from damage (*McCool and Miyamoto, 2012*; *Miyamoto, 2011*), thereby contributing to acquired resistance to DNA-damaging cytotoxic therapies. Besides NEMO (also named IKKγ) and ataxia telangiectasia mutated (ATM), two known crucial regulators of the genotoxic stress-activated NF-κB signaling pathway (*Miyamoto, 2011*), poly(ADP-ribose) polymerase 1 (PARP1) was recently revealed to be indispensable for the signaling cascade that links nuclear DNA damage recognition to cytoplasmic IKK activation (*Stilmann et al., 2009*). Sequential posttranslational modifications (PTMs), including phosphorylation, ubiquitination and SUMOylation, of these signaling regulators are critical for NF-κB activation following DNA damage (*Huang et al., 2003*; *Mabb et al., 2006*; *Wu et al., 2006*). In this regard, PARP1 is an abundant nuclear protein that senses and contributes to the repair of DNA single-strand breaks (SSBs) and of DNA double-strand breaks (DSBs), working by catalyzing poly(ADP-ribosyl)ation (PARylation) of itself, histones, and other target proteins (*Gibson and Kraus, 2012*). In particular, PARP1-catalyzed PARylation has emerged as a vital means for the rapid assembly of signaling complexes that are critical for DNA damage-initiated NF-κB activation (*Mabb et al., 2006*; *Stilmann et al., 2009*). However, the cooperative function of NF-κB with other key stress elements in cellular resistance to DNA-damaging therapies remains to be clarified. How DSBs trigger the activities of such a large number of factors with such precise spatiotemporal coordination also remains unclear. Leukemia-Related Protein 16 (LRP16), a member of the macro domain family, was identified as a PAR-binding protein in genotoxic threat-treated cells and a putative substrate of PARP1, which suggests that LRP16 could be an important molecule in the cellular response to DNA damage (*Han et al., 2011*; *Timinszky et al., 2009*). We have previously established a critical role for

LRP16 in DSB-induced activation of the NF-κB signal transduction cascade, which counteracts apoptosis and allows cells to escape the lethal effects of DNA damage (*Wu et al., 2015*). Mechanistically, LRP16 facilitates the lesion-specific recruitment of PARP-1 and NEMO through its constitutive interactions with these two proteins, and then ultimately facilitates the concomitant recruitment of ATM and protein inhibitor of activated STAT Y (PIASy) to NEMO to ensure the activation of NF-κB after the induction of DSBs (*Wu et al., 2015*). Although these studies have considerably advanced our understanding of the cellular response to DNA damage, the genotoxic stress-initiated nucleoplasmic NF-κB signaling pathway remains poorly understood, in particular, the early signaling networks linking DNA lesion recognition in the nucleus to subsequent activation of IKK and liberation of NF-κB in the cytoplasm. The double-stranded-RNA (dsRNA)-activated protein kinase PKR, a ubiquitously expressed serine/threonine kinase, has been implicated in the regulation or modulation of cell growth through multiple signaling pathways and has also been described as a signal integrator in the translational and transcriptional control pathways (*Bennett et al., 2006*; *Dar et al., 2005*; *Donzé et al., 2004*; *Liu et al., 2013*). In addition to its functional regulatory function, PKR has a role in signal transduction and transcriptional control through the IκB/NF-κB pathways (*Gil et al., 2000*, *Gil et al., 2004*). PKR has also been implicated in different stress-induced signaling pathways including dsRNA signaling to NF-κB activation, although the precise function of PKR in these signaling pathways remains controversial. Not only is PKR an effector molecule in the cellular response to dsRNA, but it also integrates signals in response to the activation of Toll-like receptors, growth factors, and diverse cellular stresses (*Hsu et al., 2004*; *Nakamura et al., 2010*). PKR is involved in multiple pro- and anti-apoptotic pathways in normal and cancer cells. Investigators have shown that PKR specifically interacts with STATs, FADD, and IKK (*Williams, 2001*), and researchers have also described a novel role for PKR as a mediator of IR resistance, modulated partly by the protective effects of NF-κB activation (*von Holzen et al., 2007*). Although some models have been proposed, the precise molecular mechanism by which PKR activates IKK/NF-κB pathways remains to be determined. Moreover, PKR is also involved in many cellular pathways, exerting various functions on cell growth and tumorigenesis (*Marchal et al., 2014*). However, the exact role of PKR in cancer biology remains controversial. Chemotherapeutic drugs such as doxorubicin, etoposide, and 5-fluorouracil are able to induce and activate PKR protein, triggering apoptosis (*García et al., 2011*; *Peidis et al., 2011*; *Yoon et al., 2009*). Conversely, PKR has been suggested to be involved in the neoplastic process of the proliferative transcription factor NF-κB (*Delgado André and De Lucca, 2007*). Curiously, different expression patterns of PKR/eIF2α/NF-κB activity, even in the same type of cancer, such as different cholangiocarcinoma cell lines, point to the complexity of the role of PKR in cancer (*Kunkeaw et al., 2013*).

In the current study, we report how DNA-damage-induced nuclear events are linked to the activation of cytoplasmic IKK kinase, thereby activating NF-κB. LRP16 selectively interacts and activates PKR, and it also acts as a scaffold to assist the formation of a ternary complex of PKR and IKKβ, prolonging PAR-dependent NF-κB transactivation caused by genotoxic threats. We also discovered a small molecule MRS2578 that could profoundly abolish these interactions, converting LRP16 into a death molecule and forestalling colon tumorigenesis and acquired resistance to DNA-damaging therapies driven by the oncogenes of the NF-κB pathway. This study takes CRC as a model to understand mechanisms that account for a limited response of genotoxic therapies in solid tumors and to seek combination solutions. Inclusion of etoposide plus MRS2578, versus each drug alone, exhibits synergistic tumor cytotoxicity both ex vivo and in vivo. Our combinatorial approach introduces a strategy to enhance the efficacy of DNA-damaging cytotoxic therapies for the treatment of cancer. Thus, targeting the biological function of LRP16 will provide the proof of principle for two understudied concepts in cancer therapy: (1) blocking subsignals, rather than total signals, as a means of impeding oncogenic NF-κB signaling and (2) targeting regulatory protein–protein interactions as a way to produce effective antitumor agents and to sensitize tumor cells to DNA-damaging cytotoxic therapies.

## Results

### Massive LRP16 expression predicts a poor prognosis in human CRC patients

To clarify the clinicopathological relevance of LRP16 in patients with CRC, we used immunohistochemistry (IHC) to examine the LRP16 protein levels in a human tissue array containing 202 CRC clinical specimens with paired adjacent normal colon tissues from patients with CRC. An analysis using the Image-Pro Plus software showed that the level of LRP16 expression was significantly higher in the CRC tissue samples than the adjacent normal tissues (*Figure 1A–B*). LRP16 was highly elevated in primary CRC tumors compared with their adjacent normal tissues as determined by reverse transcription polymerase chain reaction (RT-PCR) and Western blot analysis (*Figure 1—figure supplement 1A*). These CRC samples were staged according to the system developed by the American Joint Committee on Cancer (AJCC), also known as the Tumor-Node-Metastasis (TNM) system. Notably, we found that the level of LRP16 expression positively correlated with the histological grades of the tumors and was also strongly associated with a higher tumor stage (*Figure 1C–D* and *Figure 1—figure supplement 1A* and *Figure 1—source data 1*), suggesting that the level of LRP16 expression is progressively elevated during the progression of the patients with CRC. A Kaplan–Meier survival analysis of the expression of LRP16 and the clinical behavior of CRC further showed that a low level of LRP16 was associated with better overall survival (OS) in CRC patients (p=0.0224) (*Figure 1E*). In this context, LRP16 expression was an independent prognostic factor, with a hazard ratio (HR) of 0.58 (95% confidence interval [CI] 0.36–0.90) in a multivariate analysis (tumor grades HR 0.46 [95% CI 0.26–0.82]; tumor stages HR 0.44 [95% CI 0.21–0.93]), which is similar to our previous findings (*Xi et al., 2010*). Interrogation of the most comprehensive public database, The Cancer Genome Atlas (TCGA) (*Cancer Genome Atlas Network, 2012*), also supported the notion that the expression of LRP16 transcripts is profoundly elevated in CRC. Interestingly, this database also showed that when CRC samples were further stratified according to their tumor stage, the level of LRP16 expression also positively correlated with the clinical tumor stages (TNM stage) of these patients with CRC (*Figure 1F–G*). Further analysis of our data and the TCGA database showed that the level of LRP16 expression also positively correlated with the histological grades of these CRC patients and that, remarkably, high levels of LRP16 expression in CRC strongly correlated with lymph-node positivity and metastasis in the cohort of patients with CRC (*Figure 1H–I* and *Figure 1—source data 1*).

NF-κB is constitutively activated in many malignancies, including CRC (*Sakamoto et al., 2009*), but the molecular mechanism underlying the constitutive activation of NF-κB in tumors remains to be defined. NF-κB activation was defined as the detection of p65 nuclear staining in over 50% of the tumor cells in the CRC tissues. Constitutive activation of NF-κB was observed in 28.7% (58 of 202) of the cohort of patients with CRC (*Figure 1J*). The total p65 levels slightly differed among the CRC samples, but they were significantly elevated in CRC samples compared with the adjacent normal tissues (*Figure 1J*), and also positively correlated with the histological grades of the tumors (*Figure 1—figure supplement 1B*). An analysis of consecutive tissue sections showed that the level of LRP16 expression positively correlated with the level of p65 expression (*Figure 1K*). The correlation between LRP16 and p65 was further confirmed in a larger scale array of 202 samples. Specifically, approximately 66% of the samples with high LRP16 expression displayed high p65 expression, whereas approximately 68% of the low LRP16 samples displayed low p65 expression (*Figure 1L*). Similar to the expression of LRP16, the phosphorylation of p65 (phospho-p65) at Ser536, which represents the activated form of NF-κB, was significantly higher in CRC tissues. Analysis of phospho-p65 protein expression by Western blot analysis mirrored that of LRP16, with the highest expression observed in CRC samples. Most importantly, high expression levles of phospho-p65 correlated positively with high expression levels of LRP16 in CRC clinical specimens (*Figure 1—figure supplement 1B*). Informatively, XIAP (a target of NF-κB) expression was significantly elevated in all 202 CRC tissues compared with the adjacent normal tissues (*Figure 1—figure supplement 1C*). However, we did not observe a significant correlation between LRP16 and XIAP expression in these CRC samples (*Figure 1—figure supplement 1D*). Moreover, analysis of the TCGA CRC RNA-seq dataset revealed that *LRP16* expression was also not significantly correlated with that of *BCL2L1* (anti-apoptotic transcriptional target of NF-κB), but it was inversely correlated to that of *XIAP* (*Figure 1—figure supplement 1D*). A different trend in the relevance of LRP16 and XIAP expression in our and the TCGA

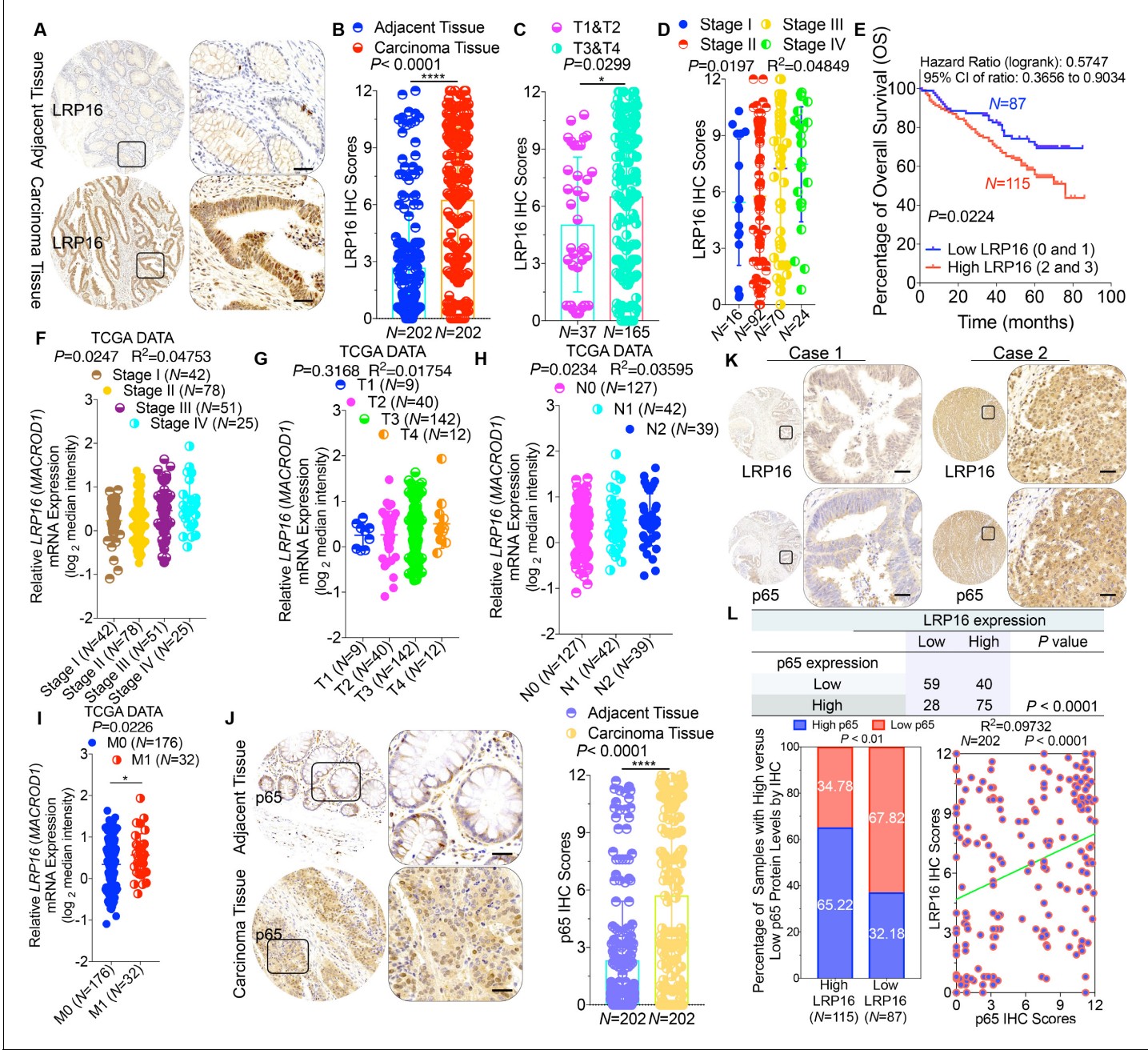

**Figure 1.** LRP16 is progressively elevated expression during progression of CRC. (**A**) Representative images of tumors and matched adjacent tissues from a tissue array containing 202 CRC samples paired with adjacent normal colon tissues, which were immunohistochemically (IHC) stained for LRP16 expression. The outlined areas in the left images are magnified on the right. Scale bars, 50 μm. (**B**) LRP16 expression scores by IHC shown as scatter dot plots. We compared the CRC tissues with the matched adjacent normal colon tissues using the Wilcoxon matched pairs test (*n* = 202). Error bars represented the mean ± SD. *p<0.05, **p<0.01, ***p<0.001, ****p<0.0001. (**C–D**) Analysis of LRP16 expression levels in CRC samples based on the indicated stratification (T stages and tumor stages). Error bars represented the mean ± SD. *p<0.05, **p<0.01, ***p<0.001, ****p<0.0001. (**E**) Kaplan–Meier plot of the overall survival (OS) of 202 patients with CRC (two groups stratified by LRP16 expression level: low LRP16 [0 and 1], *N* = 87; high LRP16 [2 and 3], *N* = 115). A log-rank test was used to compare the differences between two groups. (**F–I**) Raw data were exported from The Cancer Genome Atlas (TCGA) CRC database. Whisker plots of the expression of LRP16 in CRC samples based on the indicated stratification, T stages (**F**), tumor stages (**G**), lymph-node metastasis (**H**), distant metastasis (**I**). Data presented as the log2 median-centered ratio expression. Unpaired Mann–Whitney test was used to evaluate statistical significance. Error bars represented the mean ± SD. *p<0.05, **p<0.01, ***p<0.001, ****p<0.0001. (**J**) Representative images of CRC tissue samples and matched adjacent tissues from a tissue array containing 202 CRC samples paired with adjacent normal colon tissues, IHC stained for p65 expression. The outlined areas in the left images are magnified on the right. Scale bars, 50 μm. p65 expression scores are shown as scatter dot plots. We compared the CRC tissues with the matched adjacent normal colon tissues with the Wilcoxon matched pairs test (*N* = 202). Error

*Figure 1 continued on next page*

*Figure 1 continued*

bars represented the mean ± SD. *$p<0.05$, **$p<0.01$, ***$p<0.001$, ****$p<0.0001$. (K) Representative images of two serial sections from the same tissue samples IHC stained for LRP16 and p65. Scale bars, 50 μm. (L) Positive correlation between LRP16 and p65 expression levels. The 202 samples were classified into two groups (low LRP16, $N = 87$; high LRP16, $N = 115$) based on the LRP16 level relative to the score for the whole cohort. $p<0.01$, calculated with both the $\chi^2$ and Mann–Whitney tests. Pearson correlation between LRP16 and p65 expression levels ($N = 102$; $p<0.0001$; $R^2 = 0.09732$). The online version of this article includes the following source data and figure supplement(s) for figure 1:

**Source data 1.** LRP16 expression in tumor-adjacent tissues of the colorectal carcinoma (CRC) patient subgroups according to clinical pathological parameters.

**Figure supplement 1.** LRP16 is significantly elevated and positively correlates with constitutive NF-κB activation in human CRC samples.

cohorts of patients with CRC might be attributable to the dynamic expression patterns of NF-κB target genes, which did not follow a standardized protocol and differed in the number of CRC samples analyzed and in the patient inclusion criteria, making it difficult to compare results between groups. Taken together, these results provide overwhelming clinical evidence that LRP16 is overexpressed in human CRC samples compared with adjacent normal samples, and also confirms the critical role of LRP16 in promoting CRC tumorigenesis.

## LRP16 attenuates the cytotoxic and cytostatic effects of DNA-damaging therapeutics

To explore the biological role of LRP16 in CRC, we screened a panel of CRC cell lines for their endogenous LRP16 levels (*Figure 2A*). We also observed that the level of phospho-p65 expression was positively associated with the level of LRP16 expression (*Figure 2—figure supplement 1A*). Next, we investigated the cytotoxic and cytostatic effects of chemotherapeutic drugs on CRC cell lines in culture models. In these cells, we compared the cytotoxic and cytostatic effects of etoposide to those of 5-fluorouracil and oxaliplatin, as reflected by their half-maximal growth inhibitory concentrations ($IC_{50}$). The results indicated that CRC cell lines (i.e. LS180, HCT116, and RKO) expressing relatively higher levels of endogenous LRP16 were less sensitive to etoposide. In contrast, CRC cell lines expressing relatively lower or undetectable levels of endogenous LRP16 (i.e. SW480, CaCO2, and LoVo) were more sensitive to etoposide (*Figure 2—figure supplement 1B*). Based on our findings, it is reasonable to speculate that the molecular mechanisms underlying the therapeutic response difference to the cytotoxic and cytostatic effects of etoposide might be dependent on the different levels of LRP16 expression in CRC cell lines. We further examined whether expression of exogenous LRP16 could restore the resistance of CRC cell lines to a basic low level of LRP16 expression in response to chemotherapeutic drugs. As expected, the forced overexpression of LRP16 in SW480 cells with a basic low level of LRP16 rendered tumor cells more resistant to etoposide than to the other two drugs, as evidenced by increased cell viability and clonogenicity (*Figure 2B* and *Figure 2—figure supplement 2A–B*). Increasing the expression of LRP16 substantially rescued etoposide-induced apoptosis, as evidenced by reduced caspase three cleavage (*Figure 2C*). However, no significant protective effects were observed on the invasive and metastatic capacities of the CRC cell line ex vivo (*Figure 2—figure supplement 2C*).

Next, we assessed how LRP16-mediated anti-apoptotic signaling accounted for the limited response to etoposide. A conceivable possibility is that NF-κB activation caused by LRP16 in response to etoposide could account for this protective effect (*Figure 2D*). We found that elevated phospho-p65 levels, which were associated with substantially enhanced NF-κB transcriptional activity, occurred in the most etoposide non-responsive cell lines. Conversely, in some etoposide-responsive cells, upregulation of the phospho-p65 level or NF-κB transcriptional activity did not remarkably occur (*Figure 2—figure supplement 1B* and *Figure 2—figure supplement 2D*). The Western blot results also showed that with a gradually increase in LRP16 expression significantly increased the phosphorylated forms, but not the total forms, of the upstream regulators of the NF-κB pathway, IKKα and IKKβ, and also increased the levels of phospho-p65 (*Figure 2E*), compared with those in controls. Similar results in clonogenicity experiments demonstrated that the gradual re-expression of exogenous LRP16 in SW480 cells restored their resistance to etoposide (*Figure 2—figure supplement 3A*).

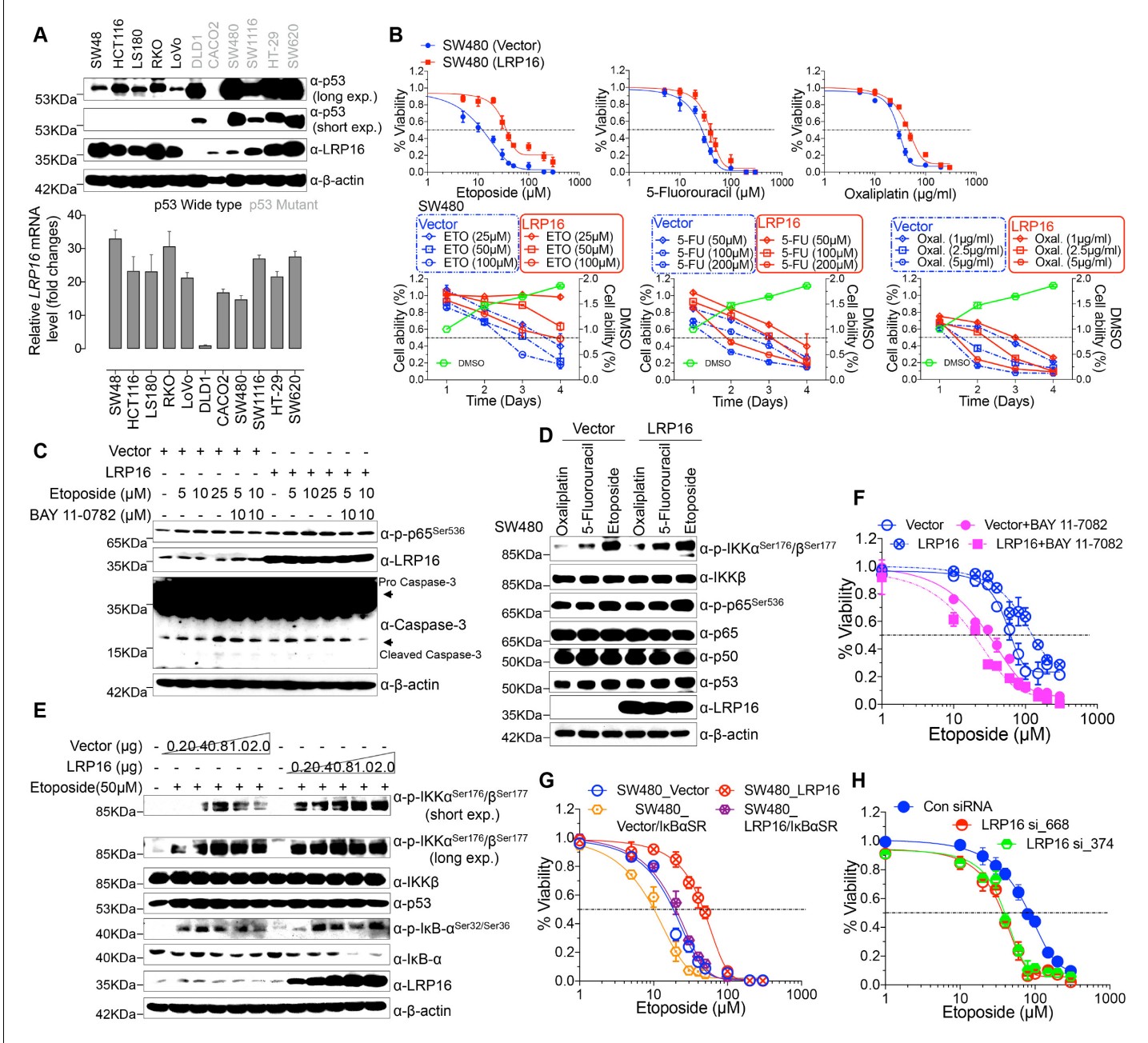

**Figure 2.** LRP16 mediates acquired resistance to etoposide in CRCs. (**A**) Expression levels of LRP16 were analyzed in 11 CRC cell lines with Western blotting and RT–qPCR. (**B**) Cell growth assay. SW480 cells were transfected with the control vector or an LRP16-expressing plasmid for 48 hr, and then treated with etoposide, 5-fluorouracil, or oxaliplatin for a further 72 hr. Cell viability was determined with a CCK-8 assay. (**C**) SW480 cells transfected with an LRP16-expressing plasmid or the control vector were treated for 36 hr with the indicated concentrations of etoposide and/or BAY 11–7082. Cell lysates were separated with SDS-PAGE and analyzed with Western blotting using the indicated antibody. β-Actin was used as the loading control. (**D**) SW480 cells transfected with an LRP16-expressing plasmid or the control vector were treated with etoposide (50 µM), 5-fluorouracil (100 µM), or oxaliplatin (2 µg/ml) for 3 hr, and the cell lysates were analyzed with Western blotting using the indicated antibodies. (**E**) SW480 cells transfected with increasing doses of an LRP16-expressing plasmid or the control vector were treated with etoposide (50 µM), and then immunoblotted with the indicated antibodies. (**F**) SW480 cells transfected with an LRP16-expressing plasmid or the control vector were pretreated with or without BAY 11–7082 and then treated with etoposide for indicated concentrations, and subjected to a cell viability analysis. (**G**) SW480 cells transfected with the indicated plasmids and treated with etoposide for indicated concentrations, and subjected to a cell viability analysis. (**H**) SW620 cells were transfected with control or LRP16-directed shRNAs, and the resulting stable cells were treated with etoposide and subjected to a cell viability analysis. Data are representative of at least three independent experiments.

The online version of this article includes the following figure supplement(s) for figure 2:

*Figure 2 continued on next page*

**Figure supplement 1.** The cytotoxic and cytostatic effects of chemotherapeutic drugs on CRC cell lines.
**Figure supplement 2.** LRP16 plays a critical protective role for the survival of CRC cells.
**Figure supplement 3.** The protective roles of LRP16 for CRC cell survival rely on PAR-dependent NF-κB activation in response to DNA damage.

Hence, we wondered whether NF-κB activation enhanced by LRP16 potentially contribute to the limited response to etoposide. To achieve this goal, we assessed the impact of the NF-κB inhibitor, BAY 11–7082, and the IκBα super-repressor, IκBαSR, a dominant-negative mutant of IκBα (*Wu et al., 2015*), which blocks the effects of NF-κB activation on the viability and clonogenicity of CRC cells. Blocking NF-κB activation with either BAY 11–7082 or IκBαSR in SW480 cells exogenously expressing LRP16 or the vector control profoundly reduced their viability and clonogenicity (*Figure 2F–G* and *Figure 2—figure supplement 3B*). These results suggest that LRP16 confers DNA damage-triggered NF-κB-mediated expression of anti-apoptotic signaling molecules and is involved in restraining the response to DNA-damaging cytotoxic therapies in CRC cells.

To determine whether LRP16 is the relevant target at the cellular level and whether cell killing is truly dependent on this particular mechanism, LRP16 was specifically knocked down using two different LRP16 small hairpin RNAs (shRNAs, siRNA_374 and _668), as described previously (*Wu et al., 2015*). Cells stably expressing *LRP16* siRNA_374, siRNA_668, or control_siRNA were treated with etoposide for 72 hr. Notably, depletion of endogenous LRP16 resulted in significantly reduced cell viability and clonogenicity and profoundly enhanced the sensitivity of cells to etoposide (*Figure 2H* and *Figure 2—figure supplement 3C*). Beyond its indispensable role in DNA repair (*Gibson and Kraus, 2012*), emerging evidence reveals that PARP1-mediated PARylation is one of the most crucial PTMs orchestrating DNA-damage-initiated NF-κB signaling (*Stilmann et al., 2009*). The high affinity of LRP16 for PAR led us to further ascertain whether LRP16 could facilitate PAR-dependent NF-κB signaling/transactivation of anti-apoptotic genes to counter intrinsic DNA damage in CRC cells and to protect against cell death induced by genotoxic agents. Of note, ectopic expression of LRP16 mutants (D160A or I161A), which was sufficient to significantly reduce its affinity for PAR, but not LRP16 wild-type (WT), in SW480 cells, dramatically sensitized the tumor cells to genotoxic stress-induced apoptosis, as conveyed by reduced cell viability and clonogenicity, in line with the indispensible role of the affinity of LRP16 for PAR in NF-κB activation and anti-apoptotic transcription (*Figure 2—figure supplement 3D*). Consistently, PARP inhibition by PJ-34 or 3-AB treatment also significantly reduced CRC cells survival and dramatically sensitized cancer cells to etoposide-induced cell death, as evidenced by the reduced cell viability and clonogenicity, in LRP16-upregulated SW480 cells (*Figure 2—figure supplement 3D*). Taken together, these data suggest that LRP16 plays a critical role in DNA-damage-initiated and PAR-dependent NF-κB transactivation, which could account for the limited or lack of response to the cytotoxic and cytostatic effects of etoposide and protection against cell death induced by genotoxic agents.

## LRP16 sustains NF-κB activation induced by genotoxic threats

To further assess whether LRP16 is an essential component of NF-κB signaling, we used an immunoblotting assay to evaluate genotoxicity-induced NF-κB transcriptional activity. The results indicated that the re-expression of LRP16 in cells exposed to etoposide profoundly increased the phosphorylated forms, but not the total forms, of the upstream regulators of the NF-κB pathway, IKKα, IKKβ and IκBα, in a dose- and time-dependent manner (*Figure 3A*). Consistent with this phenomenon, exposure to IR, together with exogenous LRP16 expression, also largely prolonged the IR-stimulated activation of NF-κB compared with that in cells transfected with control vector (*Figure 3B*). Similar results were repeated in another CRC cell line, LoVo cells (*Figure 3—figure supplement 1A–B*). Moreover, compared with control cells, etoposide treatment triggered the marked nuclear translocation of NF-κB/p65 in SW480 cells exogenously expressing LRP16 (*Figure 3—figure supplement 1C*). Conversely, LRP16 deficiency introduced by its siRNAs in cells exposed to etoposide reduced the phosphorylated forms, but not the total forms, of IKKα, IKKβ, and IκBα, in a dose- and time-dependent manner (*Figure 3C*). Consistent with these data, IR triggered the phosphorylation of IKKα, IKKβ, and p65, and IκBα was substantially reduced in SW620 cells expressing LRP16-specific siRNAs compared with those transfected with scrambled nonspecific siRNAs (*Figure 3D*). Moreover,

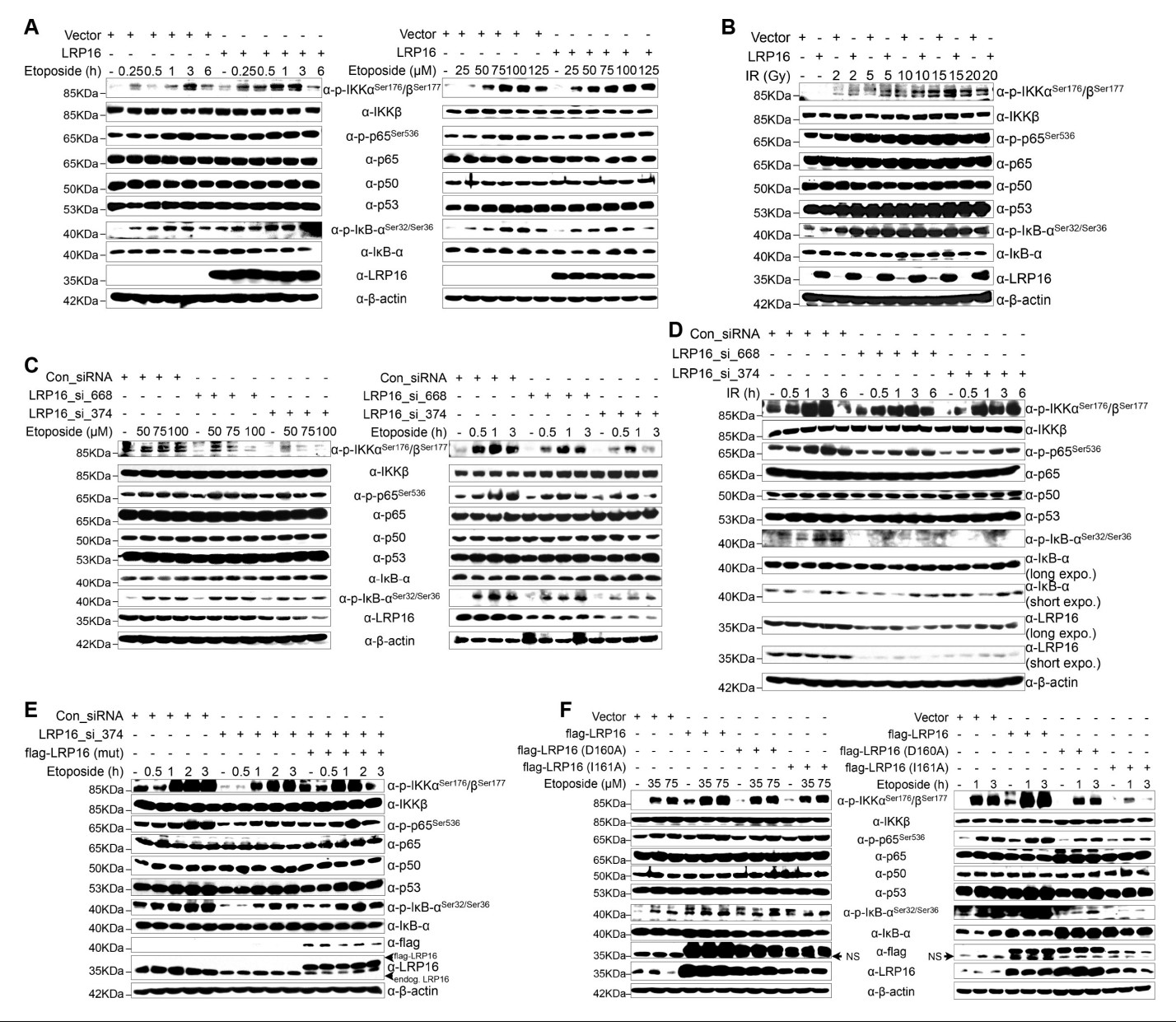

**Figure 3.** LRP16 is required for genotoxicity-induced NF-κB activation. (**A**) Whole cell lysates from cells transfected with an LRP16-expressing plasmid or the control vector were treated with the indicated concentrations of etoposide for 2 hr or with 50 μM etoposide for the indicated periods, and then immunoblotted with the indicated antibody. β-Actin was used as the loading control. (**B**) SW480 cells transfected with an LRP16-expressing plasmid or the control vector were treated with IR for the indicated dose. The cell lysates were then separated with SDS-PAGE and analyzed with Western blotting using the indicated antibody. (**C**) Whole-cell lysates from SW620 cells transfected with the indicated siRNAs and treated with the indicated concentrations of etoposide for 2 hr or with 50 μM etoposide for the indicated periods were immunoblotted with the indicated antibody. β-Actin was used as the loading control. (**D**) SW620 cells were transfected with the indicated siRNAs and exposed to IR for the indicated periods. Their lysates were separated with SDS-PAGE and analyzed with Western blotting using the indicated antibody. (**E**) SW620 cells were cotransfected with the indicated siRNAs and/or the LRP16-expressing vector that contained silent mutations in the sequences that targeted by the LRP16 siRNAs, and were then treated with 50 μM etoposide for the indicated periods. The cell lysates were immunoblotted using the indicated antibody. (**F**) Whole cell lysates of cells transfected with the control vector or vectors expressing LRP16 or the LRP16 mutants (LRP16_D160A, LRP16_I161A) and treated with the indicated concentrations of etoposide for 2 hr or with 50 μM etoposide for the indicated periods were immunoblotted with the indicated antibody. β-Actin was used as the loading control. One representative experiment of three was shown.

The online version of this article includes the following figure supplement(s) for figure 3:

**Figure supplement 1.** LRP16 executes an essential function in the nuclear-initiated NF-κB signaling in response to DNA damage.

**Figure supplement 2.** LRP16 is required for NF-κB transactivation in response to genotoxic stress.

*Figure 3 continued on next page*

Figure 3 continued

**Figure supplement 3.** LRP16 is not critical for NF-κB1 and TAK1 activation during the cellular response to gentoxic stress.

**Figure supplement 4.** Ectopic of LRP16 augments DNA-damage-triggered NF-κB-mediated expression of anti-apoptotic molecules.

the nuclear accumulation of NF-κB/p65 was also substantially reduced in the LRP16-specific siRNAs-transfected cell lysates compared with that in the controls (*Figure 3—figure supplement 1D*). Taken together, these results support the critical function of LRP16 in controlling the DNA damage-stimulated activation of NF-κB.

To control for potential off-target effects of the LRP16 siRNA and to confirm that LRP16 deficiency alone impairs the genotoxicity-induced activation of NF-κB, we generated an LRP16-expressing vector that contained silent mutations in the sequences that were targeted by the LRP16-directed siRNAs. Our results showed that the etoposide-triggered phosphorylation of IKKα, IKKβ, p65, and IκBα was substantially reduced in the LRP16 siRNA_374-transfected cell lysates. Conversely, exogenous expression of LRP16 with a silent mutation at the shRNA target site rescued this effect (*Figure 3E*). However, our results also indicated that, compared with the control, the exogenous expression of LRP16 had almost no effect on the activation of NF-κB1 (noncanonical NF-κB signaling pathway) induced by etoposide (*Figure 3—figure supplement 2A*). These data suggest that LRP16 is a key regulator of the activating phosphorylation of the activation loops of IKKs and hence of IKK function in the physiological context. Exactly how any specific signaling pathway targets the IKK complex remains contentious. TAK1 is considered to be the immediate upstream activator of IKK and an essential component of both the nuclear and receptor-mediated activation of NF-κB, phosphorylating the activation loops of the IKKs (*Perkins, 2007*). Next, we asked whether LRP16 regulates the kinase activity of TAK1 upon DNA damage and mediates activation of the IKK complex. Unexpectedly, the exogenous expression of LRP16 in SW480 cells did not induce significant phosphorylation of TAK1 compared with that in the control (*Figure 3—figure supplement 2B*), raising the additional possibility that other upstream kinases could contribute to the activation of IKK mediated by LRP16 within the context of etoposide-induced NF-κB signal transduction.

Next, we sought to evaluate whether the function of LRP16 in genotoxic stresses-induced NF-κB activation was dependent on its PAR binding ability. Ectopic expression of LRP16 WT, but not its mutants (D160A or I161A), which were sufficient to significantly reduce its affinity to PAR (*Wu et al., 2015*), significantly enhanced NF-κB activation after etoposide treatment, as conveyed by the phosphorylated forms of IKKα and IKKβ, p65, and IκBα (*Figure 3F*). A luciferase assay using an NF-κB-responsive element demonstrated that NF-κB transcriptional activity is altered in response to the modulation of LRP16 upon stimulation of DNA damage. Ectopic of LRP16 remarkably enhanced the levels of NF-κB-dependent luciferase reporter gene activity in both SW480 and LoVo cells following either etoposide or IR treatment (*Figure 3—figure supplement 3A*). Conversely, LRP16 deficiency introduced by two siRNAs in both HCT116 and SW620 cells considerably diminished the NF-κB transcriptional activity in response to etoposide or IR (*Figure 3—figure supplement 3B*). Of note, ectopic expression of LRP16 WT, but not its mutants (D160A or I161A) dramatically enhanced the NF-κB transcriptional activity in response to etoposide or IR (*Figure 3—figure supplement 3A*), thus supporting the critical function of LRP16 in controlling DNA damage-initiated and PAR-dependent NF-κB signaling activation.

NF-κB-mediated transcription of a panel of anti-apoptotic molecules is an important factor for cell fate determination after DNA damage. To further gain support for the notion that LRP16 regulates distinct target genes of NF-κB, an exploratory microarray analysis was performed. The results indicate that cells exogenously expressing LRP16 in response to etoposide develop alterations of the global transcription profile (*Figure 3—figure supplement 4A*). Further pathway analysis has revealed several pathways that are highly affected by LRP16, including protein-K63-linked ubiquitination, IκB phosphorylation, and the DNA damage response (*Figure 3—figure supplement 4B*). To further confirm that LRP16 is involved in the expression of NF-κB-dependent genes, human NF-κB signaling pathway PCR array was used. The results indicate that after exposure to etoposide, the expression of apoptosis-related genes was markedly downregulated, whereas the expression of anti-apoptosis-related genes was strikingly increased in cells with exogenously expression of LRP16, compared with the control cells (*Figure 3—figure supplement 4C*). These results suggest that LRP16 is

functionally involved in the NF-κB-dependent gene expression induced by etoposide stimulation and might account for the compromised therapeutic efficacy of etoposide.

## LRP16 physically interacts with PKR and IKKβ in the cytoplasm

We hypothesized that a unique LRP16-containing protein complex forms under genotoxic threat conditions and that the complex component(s) affect NF-κB activity. Thus, LRP16 interactors were purified by immunoaffinity purification (*Nakatani and Ogryzko, 2003*), and we identified the LRP16-associated proteins by exhaustive rounds of 'shotgun' liquid chromatography and high-throughput mass spectrometry (LC–MS/MS) (*Figure 4A*). An in-depth bioinformatics analysis of the LC–MS/MS data indicated that PKR with the matching peptide potentially interacted with LRP16. Importantly, the interaction of endogenous LRP16 with PKR protein was confirmed by coimmunoprecipitation (co-IP) in SW480 cells after etoposide treatment (*Figure 4B*). To exclude the possibility that the interaction between LRP16 and the PKR protein was indirect and mediated by DNA or chromatin, ethidium bromide (EB) was added to the cell lysates during co-IP. As expected, ethidium bromide did not markedly impair the etoposide-induced interaction between LRP16 and PKR, suggesting that this interaction was also DNA-independent (*Figure 4C*).

To identify the region within PKR responsible for its interaction with LRP16, we generated several PKR deletions and mutant constructs, and co-expressed them with LRP16 in SW480 cells. A deletion analysis demonstrated that the C-terminal catalytic domain of PKR mediated its physical interaction with LRP16 (*Figure 4D*). To map the regions in LRP16 that interacted with PKR, we generated a series of LRP16 deletion mutant constructs and co-expressed them with PKR in SW480 cells. *Figure 4E* shows that the macrodomain of LRP16 retained is ability to interact with PKR.

Next, we performed a glutathione S-transferase (GST) pull-down assay using GST-fused LRP16 and in vitro-transcribed and -translated components of WT PKR and PKR deletions. Our pull-down assays using the recombinant proteins demonstrated a direct LRP16–PKR interaction (*Figure 4F*). Similarly, in vitro GST pull-down assays indicated that almost all PKR fragments, except ΔPKc, directly bound to LRP16. In contrast to the strong association between LRP16 and full-length PKR, the ΔPKc-truncated PKR protein interacted negligibly with LRP16 (*Figure 4G*), further supporting the critical role of the C-terminus of PKR in the LRP16–PKR interaction. Intriguingly, according to our LC–MS/MS data, IKKβ with two matching peptides potentially interacted with LRP16 (*Figure 4A*). Consistent with this finding, previous studies have shown that PKR binds specifically to the IKKβ subunit of the IKK complex and mediates the activation of NF-κB (*Bonnet et al., 2000*; *Zamanian-Daryoush et al., 2000*). To investigate whether the physical associations between LRP16 and PKR or IKKβ reflected a capacity of LRP16 to interact with both the PKR and IKKβ proteins simultaneously, or whether LRP16 interacted with either PKR or IKKβ in different cellular environments, a GST pull-down assay was performed using GST-fused LRP16 and in vitro-transcribed and -translated components of PKR and IKKβ. The assay indicated that LRP16 was capable of interacting directly with both PKR and IKKβ, but independently with each (*Figure 4H*). Together, these experiments not only showed the molecular details involved in the interaction between LRP16 and PKR, but also provided additional support for the physical associations among LRP16, PKR, and IKKβ in vitro.

We further identified the contribution of LRP16 to the assembly of this PKR–IKK kinase complex. Notably, depletion of LRP16 introduced by its siRNAs resulted in the significant attenuation of the PKR–IKKβ interaction, both at baseline and after stimulation with etoposide (*Figure 4I*), suggesting that LRP16 was required for the formation of a ternary complex with PKR and IKKβ. The link between PKR and TP53 has been described most frequently in cancer cells, in which there is a bidirectional and complex regulatory relationship between the two proteins (*Cuddihy et al., 1999*). Unexpectedly, the interaction of endogenous LRP16 with the PKR protein was confirmed after etoposide treatment in TP53-null cells (*Figure 4—figure supplement 1A*), suggesting that the interaction between the two proteins did not depend on the presence of p53 under these conditions.

To understand how LRP16 drives NF-κB activity in association with the cytoplasmic proteins IKKβ and PKR, we used confocal microscopy to observe the intracellular distribution of LRP16 during etoposide, camptothecin, doxorubicin or IR stimulation. After stimulation, both endogenous and exogenous LRP16 dispersed from the nucleus to the cytosol (*Figure 4—figure supplement 1B–C*). We also prepared nuclear and cytoplasmic fractions from cells treated with etoposide or IR for various times, and examined each for the presence of LRP16. These cells also revealed the dynamic dispersion of LRP16 from the nucleus to the cytosol (*Figure 4—figure supplement 1B–C*). More

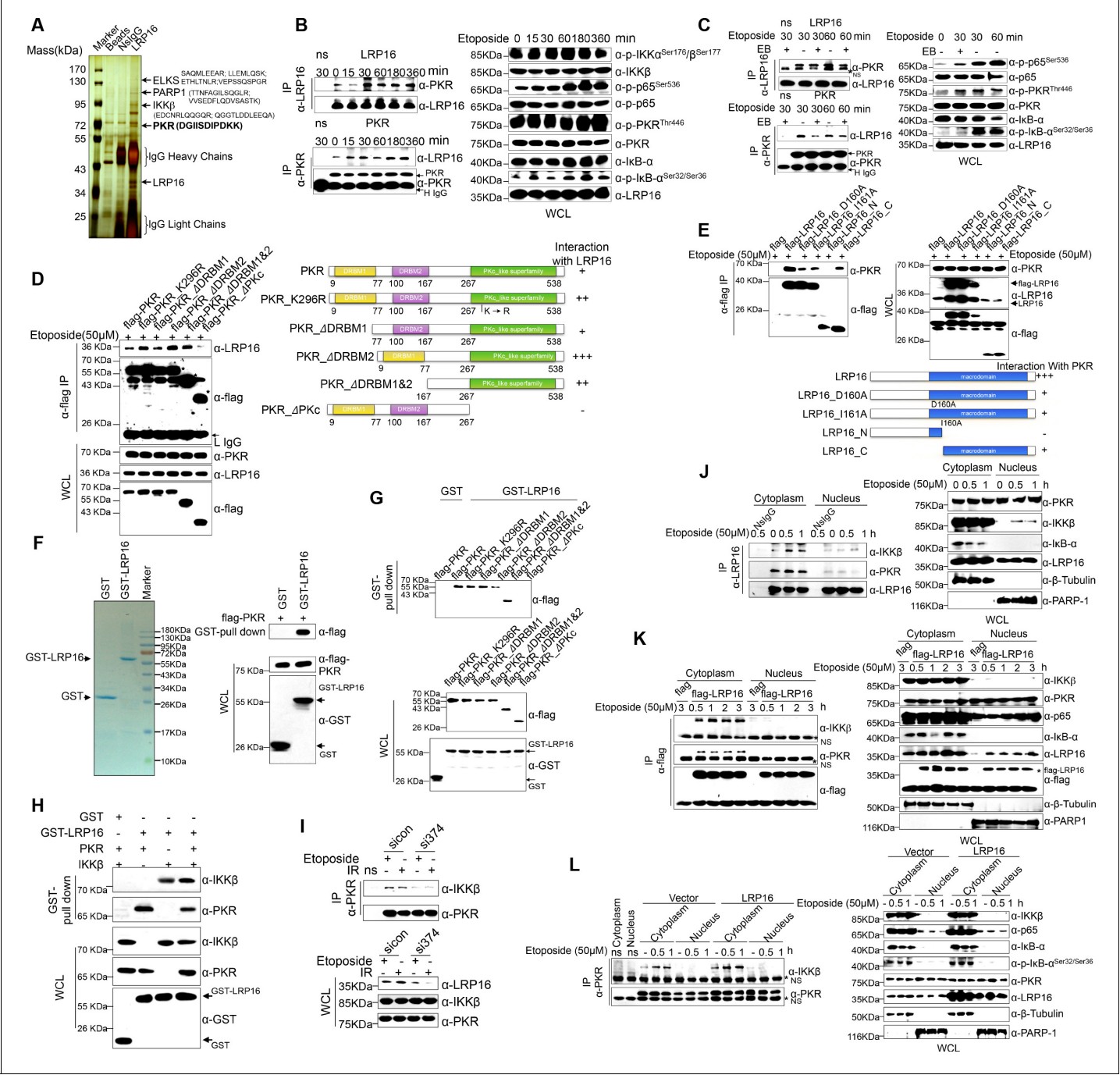

**Figure 4.** LRP16 physically interacts with PKR/IKKβ in the cytoplasm. (**A**) Mass spectrometry (MS) analysis of LRP16-associated proteins. Total cell lysate of SW620 cells was subjected to affinity purification with an anti-LRP16 antibody. The purified protein complex was resolved with SDS-PAGE and silver stained. The bands were excised, purified, and analyzed with MS. (**B–C**) Interactions of endogenous LRP16 with PKR were determined with a co-immunoprecipitated (co-IP) analysis in SW480 cells treated with the indicated periods of etoposide. (**D**) SW480 cells transfected with the indicated FLAG-tagged PKR deletion mutants and treated with etoposide for 30 min were subjected to immunoprecipitation with an anti-FLAG antibody. The lysates and immunoprecipitates were analyzed. (**E**) Cells transfected with the indicated constructs were subjected to immunoprecipitation with an anti-FLAG antibody. The lysates and immunoprecipitates were then blotted. (**F**) Bacterially expressed glutathione S-transferase (GST)–LRP16 fusion protein was purified on glutathione agarose beads; 10% of the beads were analyzed with SDS-PAGE and Coomassie Brilliant Blue staining. The remaining beads containing the GST–LRP16 fusion protein were used to pull down in vitro-translated FLAG–PKR. The interactions were detected with SDS-PAGE and immunoblotting. (**G**) In vitro-translated FLAG–PKR deletion mutants were incubated with GST or GST–LRP16–GSH–Sepharose. Proteins retained on the Sepharose were blotted with the indicated antibodies. (**H**) In vitro-translated PKR and IKKβ were incubated with GST or GST–LRP16–GSH–Sepharose. Proteins retained on the Sepharose were blotted with the indicated antibodies. (**I**) SW620 cells transfected with the indicated siRNAs were

*Figure 4 continued on next page*

*Figure 4 continued*

treated with etoposide before collection. PKR was immunoprecipitated with an anti-PKR antibody and immunoblotted with the indicated antibodies. (J–K) Cytosolic and nuclear fractions derived from SW620 cells (J), and cytosolic and nuclear fractions derived from SW480 cells expressing exogenous FLAG–LRP16 (K), were treated with etoposide (50 µM) for the indicated periods, and immunoprecipitated and immunoblotted with the indicated antibody. (L) Cytosolic and nuclear fractions derived from SW480 cells transfected with vector expressing FLAG–LRP16 or the control vector were treated with etoposide (50 µM) for the indicated periods, and immunoprecipitated and immunoblotted with the indicated antibody. One representative experiment of three was shown.

The online version of this article includes the following figure supplement(s) for figure 4:

**Figure supplement 1.** LRP16 physically interacts with PKR in a TP53 independent manner, and nucleocytoplasmic shuttling of LRP16 in response to DNA damage, not TNFα signaling depends on PAR.

importantly, the translocation of LRP16 was also detected after genotoxic stress in other tumor cells, including breast cancer and lung cancer cells (*Figure 4—figure supplement 1D*). The kinetics of LRP16 translocation were consistent with the dynamic changes in both IKKα and KKβ phosphorylation in response to etoposide stimulation, suggesting that LRP16 was possibly required for the assembly and activation of the IKK complexes triggered by DNA damage. To test this possibility, we prepared nuclear and cytoplasmic fractions from cells treated with etoposide for different times and used co-IP to pull down both endogenous and exogenous LRP16 from the control cells and cells expressing FLAG-tagged LRP16 in the presence or absence of etoposide. As shown in *Figure 4J and K*, in the etoposide-treated cells, the interactions of both endogenous and exogenous LRP16 with PKR and IKKβ were observed and confirmed by co-IP only in cytoplasmic fractions, but they were negligible in the nuclear fractions. Notably, the overexpression of FLAG-tagged LRP16 in SW480 cells upon exposure to etoposide increased the interaction between PKR and IKKβ detected with co-IP, but only in the cytoplasmic fractions, with negligible effects in the nuclear fractions (*Figure 4L*). This finding suggests that LRP16 plays an important role in the recruitment of PKR and IKKβ into a physical complex that facilitates the downstream signaling of NF-κB.

We then asked whether the intracellular trafficking of LRP16 occurs in response to stimulation with TNFα. In contrast to the etoposide-induced translocation of LRP16 to the cytoplasm, the translocation of LRP16 was completely abrogated by stimulation with TNFα (*Figure 4—figure supplement 1E*). The IR and/or etoposide-induced shift of LRP16 into the cytoplasmic complex fractions was completely blocked both by PARP inhibitors (3-AB or PJ-34) and by the PAR-binding-deficient mutant LRP16_I161A (*Figure 4—figure supplement 1F*), suggesting that this process strictly depends on the presence of PAR and the intact PAR-binding motifs in LRP16. Our data suggest that upon the detection of DNA lesions and PAR, LRP16 assembles the PKR and IKKβ complex in the cytoplasm. This complex could be formed by direct protein–PAR interactions, as well as by protein–protein interactions. The PAR chains bound to LRP16 act as an interaction platform, which is required for PKR-mediated IKK activation and the activation of the NF-κB pathway in response to DNA damage.

## Crosstalk between LRP16 and PKR is critical for genotoxicity-induced NF-κB activation

We were particularly interested in how LRP16 drives the activation of the IKK kinases by PKR. We found that PKR is ubiquitously expressed in CRC cell lines (*Figure 5—figure supplement 1A*). However, the exact role of PKR in cancer biology remains controversial, and thus we ascertained the PKR protein levels with IHC using a human tissue array containing 202 CRC samples with paired adjacent normal colon tissues. Analysis of these data showed that the level of PKR expression was significantly elevated in the carcinoma tissues relative to that in the adjacent tissues (*Figure 5A*). Moreover, PKR was highly elevated in primary CRC tumors compared with their adjacent normal tissues as determined by RT-PCR and Western blot analysis and, noticeably, also positively correlated with the histological grades of the CRC patients (*Figure 5—figure supplement 1A*), which is consistent with the previous study (*Kim et al., 2002*). Further analysis of consecutive tissue sections showed that LRP16 expression positively correlated with PKR expression. Specifically, approximately 75% of the samples with high LRP16 expression also displayed high PKR expression, and approximately 58% of the low LRP16 samples displayed low PKR expression (*Figure 5B*).

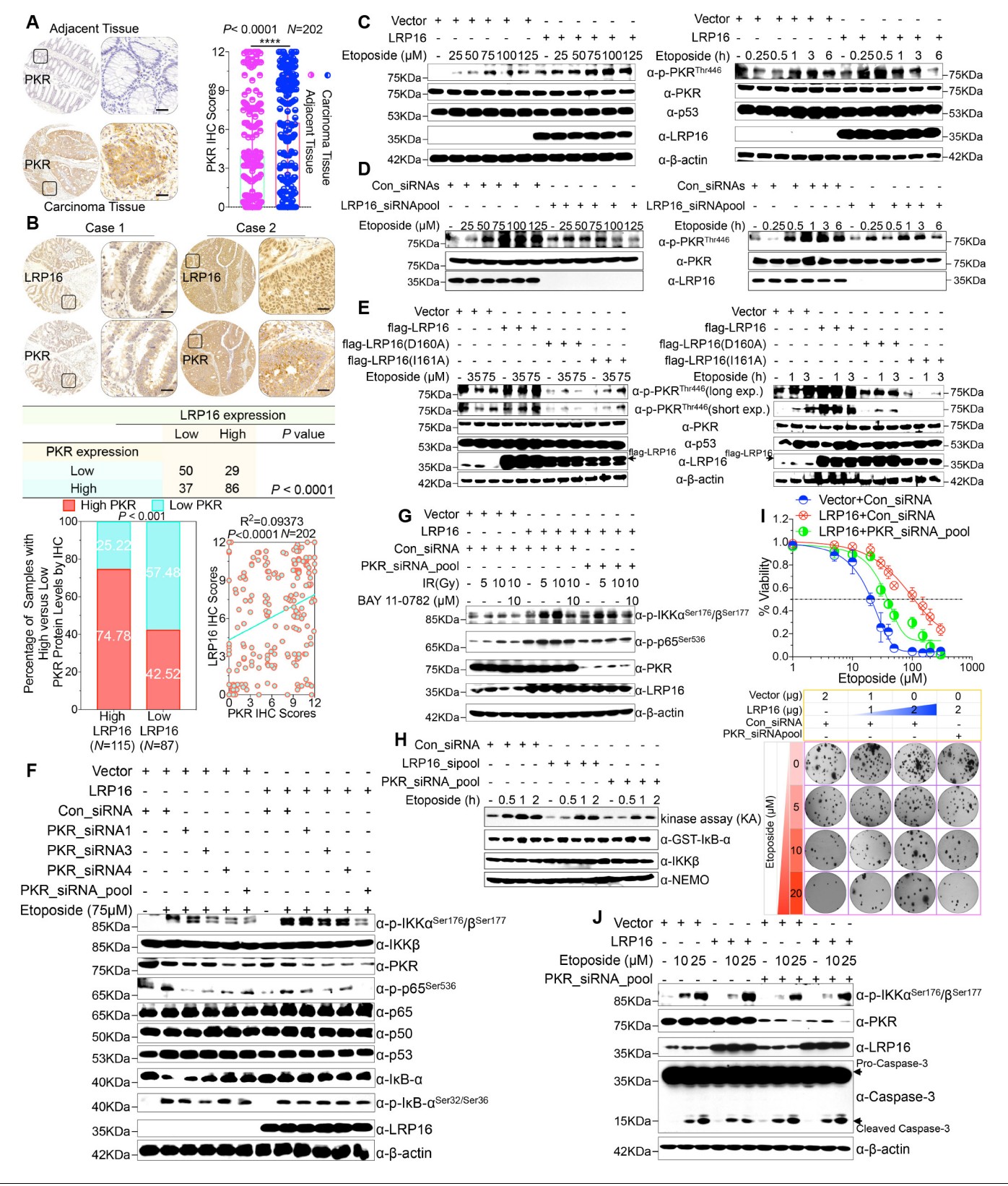

**Figure 5.** LRP16 PAR-dependently activates PKR and confers NF-κB activation induced by DNA damage. (**A**) Representative images of the IHC staining of PKR expression in CRC tissues and matched adajcent tissues from a tissue array containing 202 CRC samples paired with adjacent normal colon
*Figure 5 continued on next page*

*Figure 5 continued*

tissues. The outlined areas in the left images are magnified on the right. Scale bars, 50 µm. PKR expression scores are shown as scatter dot plots. We compared the CRC tissues with the matched adjacent normal colon tissues using a Wilcoxon matched pairs test. $N$ = 202. Error bars represented the mean ± SD. *p<0.05, **p<0.01, ***p<0.001, ****p<0.0001. (B) Representative images of IHC staining for LRP16 and PKR in two serial sections from the same tissue samples. Scale bars, 50 µm. Positive correlation between LRP16 and PKR protein levels. The 202 samples were classified into two groups (low LRP16, $N$ = 87; high LRP16, $N$ = 115) based on the LRP16 level relative to the score for the whole cohort. p<0.001, calculated by both $\chi^2$ and Mann–Whitney tests. Pearson's correlation between LRP16 and PKR expression levels ($N$ = 202; p<0.0001; $R^2$ = 0.09373). (C) Whole cell lysates from SW480 cells transfected with an LRP16-expressing plasmid or control vector were treated with the indicated concentrations of etoposide for 2 hr or with 50 µM etoposide for the indicated periods, and immunoblotted with the indicated antibody. β-Actin was used as the loading control. (D) Lysates from cells transfected with LRP16 siRNAs or control siRNAs and treated with the indicated concentrations of etoposide for 2 hr or with 50 µM etoposide for the indicated periods were immunoblotted with the indicated antibody. β-Actin was used as the loading control. (E) Cells were transfected with control vector or vector expressing LRP16 or the LRP16 mutants (LRP16_D160A or LRP16_I161A) and treated with the indicated concentrations of etoposide for 2 hr (left) or etoposide (50 µM) for the indicated periods (right). The cell lysates were immunoblotted with the indicated antibody. (F) SW480 cells either overexpressing LRP16 or transfected with the control vector were cotransfected with the indicated PKR siRNAs and treated with etoposide for 2 hr. Their lysates were immunoblotted with the indicated antibody. β-Actin was used as the loading control. (G) Cells were co-transfected with the indicated plasmids and/or siRNAs and were treated with IR at the indicated doses, with or without subsequent BAY 11–7082 treatment. Their lysates were immunoblotted with the indicated antibody. (H) In vitro kinase assay. SW620 cells were mocked transfected or transfected with the indicated siRNAs, treated with etoposide for the indicated periods, and then analyzed with an IKK kinase assay using GST–IκBα as the substrate. (I) Cells were cotransfected with the indicated plasmids and/or siRNAs, treated with etoposide, and subjected to cell viability analysis and clonogenic cell survival assays. (J) Cells were co-transfected with the indicated plasmids and/or siRNAs and treated with etoposide at the indicated concentrations. Their lysates were separated with SDS-PAGE and analyzed with Western blotting using the indicated antibody. β-Actin was used as the loading control. One representative experiment of three was shown.

The online version of this article includes the following figure supplement(s) for figure 5:

**Figure supplement 1.** PKR is elevated in CRC samples and is activated in response to DNA damage, and intact PKR, but not its kinase activity is required for the DNA damage-induced NF-κB activity.

Chemotherapeutic drugs, especially etoposide, activated PKR in CRC cell lines (*Figure 5—figure supplement 1B–C*), although the molecular mechanism that underlies PKR activation remains to be clarified. We then asked whether LRP16 affects the PKR activity induced by etoposide. Cells ectopically expressing LRP16 profoundly enhanced the PKR activity induced by etoposide in a time- and dose dependent manner (*Figure 5C*). Conversely, LRP16 depletion introduced by its siRNAs dramatically blocked the etoposide-induced activation of PKR (*Figure 5D*). We then examined whether the PAR-binding ability of LRP16 is also critical for PKR activation induced by DNA damage. Upon exposure to etoposide, the introduction of LRP16 WT, but not its mutants LRP16_D160A or LRP16_I161A, profoundly increased the activation of PKR, as evidenced by the level of its autophosphorylation, without affecting its total protein level (*Figure 5E*), suggesting that the PAR-binding activity of LRP16 is also required for the DNA-damage-induced PKR activity.

Next, we hypothesized that the interaction between LRP16 and IKKβ and NF-κB activity is required for or dependent on the activation of PKR. Western blot analysis showed that knockdown of PKR introduced by its siRNAs reduced the phosphorylated forms, but not the total forms, of the upstream regulators of the NF-κB pathway, IKKα and IKKβ, which are induced by LRP16 upon exposure to etoposide (*Figure 5F*), and also impaired the interaction between LRP16 and IKKβ after etoposide treatment (*Figure 5—figure supplement 1D*), suggesting that PKR acts as an adaptor protein and may be critical for the LRP16-mediated activation of IKK in response to DNA damage. Similarly, upon IR stimulation, depletion of PKR significantly abrogated the activation of NF-κB mediated by LRP16 (*Figure 5G*). To determine whether both LRP16- and PKR-dependent signaling to NF-κB was channeled by IKK, we used siRNAs targeting either LRP16 or PKR to repress their expression, and performed a kinase array analysis in vitro. Noticeably, the etoposide-induced activation of IKK kinases was significantly reduced in the absence of either LRP16 or PKR expression (*Figure 5H*), suggesting that the formation and integration of the ternary complex containing LRP16, PKR, and IKK kinases plays a crucial role in the DNA-damage-induced transactivation of NF-κB. To investigate the physiological role and the influence of PKR on the cellular behavior of CRC cells, we analyzed the effect of a loss-of-function PKR on cell survival and proliferation after etoposide treatment. The results of cell viability and clonogenicity assays showed that the effect of LRP16 overexpression on the resistance of CRC cells to etoposide was offset, at least in part, when PKR was simultaneously depleted (*Figure 5I* and *Figure 5—figure supplement 1E*). Similarly, knocking down PKR in cells

expressing LRP16 re-sensitized the cells to etoposide-induced DNA damage, when measured by Annexin-V/PI binding assay. Meanwhile, this phenomenon was accompanied by elevated level of caspase three cleavage (*Figure 5J* and *Figure 5—figure supplement 1F*).

Whether NF-κB activation requires the catalytic activity of PKR is still contentious (*Bonnet et al., 2000*; *Ishii et al., 2001*; *Zamanian-Daryoush et al., 2000*). Thus, we next ascertained whether the catalytic activity of PKR is required for the LRP16-mediated activation of IKK complexes during genotoxic stress. Notably, the inhibition of PKR catalytic activity with the chemical inhibitor 2-aminopurine (2-AP) had no significant effect on the activation of IKK induced by LRP16 after DNA damage (e.g. etoposide or IR) (*Figure 5—figure supplement 1G–H*), but it markedly inhibited the phosphorylation of IκBα (*Figure 5—figure supplement 1G*), which is consistent with a previous study that suggested that PKR directly phosphorylates IκBs (*Kumar et al., 1994*). These data together demonstrate that PKR activates IKK via a direct protein–protein interaction, possibly via the IKKβ subunit, rather than by its kinase activity. Thus, LRP16 activates the NF-κB signaling pathway via its interaction with and activation of PKR, which acts as an adaptor protein and activates IKK complexes in response to DNA-damaging cytotoxic therapies.

## Screening of small molecules targeting the macrodomain of LRP16

Our aforementioned results favor a model in which LRP16-mediated PKR/NF-κB activation via protein interactions limits the response of CRC cells to etoposide. We were particularly interested in the implications of this for the development of strategies to optimize the therapeutic benefits of DNA damage in CRC. Therefore, we identified small molecules that can inhibit the LRP16-mediated activation of the PKR/NF-κB pathway and small molecule inhibitors that suppress the proliferation of CRC both ex vivo and in vivo.

Several chemical libraries containing hundreds of compounds with different structures were screened, and their ability to dock with the structural pocket of the macro domain region of LRP16 (amino acids [aa] 153–319; PDB ID code: 2 × 47) was examined using the University of California, San Francisco (UCSF) DOCK 6.1 program suite (*Figure 6—figure supplement 1A*). The small molecules were ranked according to their energy scores. Among these small molecules, two compounds, MRS2578 ($C_{20}H_{20}N_5S_4$, molecular weight [MW] 472.67), a potent P2Y6 receptor antagonist (*Syhr et al., 2014*), and NECA ($C_{12}H_{16}N_6O_4$, MW 308.29), a potent adenosine receptor agonist (*Ye et al., 2016*), were identified as competitive inhibitors of the affinity of LRP16 for PAR (*Figure 6—figure supplement 1B*). In this context, we have previously shown that the PAR-binding ability of LRP16 plays an essential role as a spatial regulator of PKR/NF-κB signals by orchestrating the extranuclear signaling of IKK. Accordingly, we found that IKKβ kinase activity was markedly attenuated in IR-stimulated cells pretreated with these two compounds or the PARP inhibitors 3-AB and PJ-34 (used as positive controls) compared with that in IR-stimulated control cells. However, the inhibition of IKKβ kinase activity by NECA was less efficient than MRS2578 (*Figure 6—figure supplement 1B*). Therefore, the small molecule MRS2578 conformed to our requirements and limited the activation of NF-κB induced by DNA damage.

## MRS2578 induces cell killing and abrogates LRP16-mediated NF-κB activation in response to etoposide

To investigate the antitumor effect of MRS2578 in CRC cells ex vivo, CCK-8 assays were conducted to assess the growth of six CRC cell lines (RKO, LS180, HCT116, SW1116, SW480, and LoVo) after treatment with MRS2578. As indicated in *Figure 6A*, MRS2578 treatment for 24 hr markedly and dose-dependently reduced the viability of the CRC cell lines. Consistent with this finding, a significantly lower percentage of 5-ethynyl-2'-deoxyuridine (EdU)-incorporated (proliferating) cells was observed in the MRS2578-treated cells than in the control cells (*Figure 6B*). MRS2578 also markedly suppressed the proliferation in the CRC cells, as evidenced by their reduced clonogenic survival (*Figure 6C* and *Figure 6—figure supplement 1C*). Collectively, these data show that MRS2578 inhibits the survival and proliferation of CRC cells in vitro.

We next wondered whether MRS2578 could kill CRC cells by inactivating the LRP16-mediated activation of NF-κB induced by etoposide, and thus sensitizing the tumor cells to DNA-damaging cytotoxic therapies. Of note, etoposide-induced the activation of NF-κB, as judged by the phosphorylation forms of IKKα and IKKβ, p65, and IκBα, but not the total form, was significantly and dose-

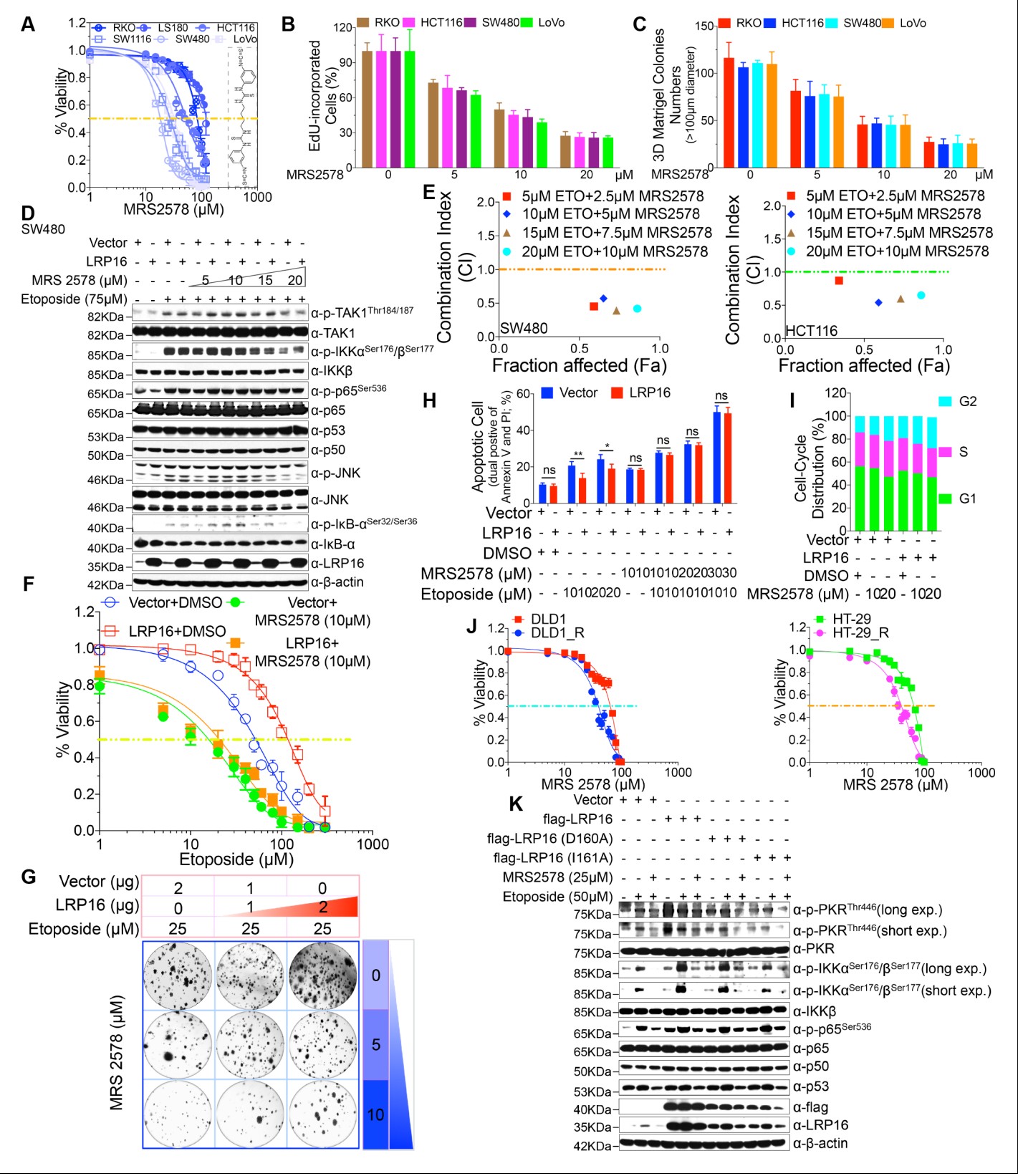

**Figure 6.** MRS2578 impairs LRP16-mediated NF-κB activity induced by DNA damage, inhibits cells proliferation, and enhances the cytotoxic effects of etoposide. (**A**) MRS2578 inhibits CRC cell viability. Cell viability was measured with the CCK-8 assay in RKO, LS180, HCT116, SW1116, SW480, and LoVo cell lines treated with the indicated concentrations of MRS2578 for 72 hr. (**B**) MRS2578 inhibited the proliferation of CRC cell lines, measured with EdU

Figure 6 continued

labeling, when the cells were treated with the indicated concentrations of MRS2578. (C) Colony formation assays: MRS 2578 suppressed colony formation in several CRC cell lines. Cells were cultured with the indicated concentrations of MRS2578 for 10 days. (D) SW480 cells transfected with the LRP16-overexpressing vector or control vector were pretreated with increasing amounts of MRS2578 for 90 min, and then with etoposide for 2 hr. Their lysates were immunoblotted with the indicated antibody. β-Actin was used as the loading control. (E) SW480 cells and HCT116 cells were treated with dual etoposide (5–20 μM) and MRS2578 (2.5–10 μM). The cells were treated daily for 3 days with the indicated drug combinations, and then assayed with CCK-8 to determine the cytotoxicity of the treatments. X axis, fraction of cells affected (Fa); y axis, combination index (CI). Combinations with CI <1 are synergistic. (F–H) Cells transfected with the control vector or a plasmid expressing LRP16 were treated with etoposide alone or in the presence of MRS2578, and subjected to cell viability analysis (F), a clonogenic cell survival assay (G), and cell apoptosis analysis (H), Error bars in (H) represented the mean ± SD for triplicate experiments. *p<0.05, **p<0.01, ns, not significant. (I) Cell-cycle analysis: Flow-Cytometric analysis of the cell-cycle profiles of SW480 cells overexpressing LRP16 or transfected with the control vector in the presence of MRS2578. (J) Cell viability assay: two CRC cell lines (DLD1 and HT-29) were exposed over time to IR until resistance emerged, generating the two cell lines DLD1_R (parental DLD1) and HT-29_R (parental HT-29). Cell viability was measured with the CCK-8 assay in the presence of MRS2578. (K) SW480 cells transfected with the control vector or with LRP16-expressing vector or the LRP16 mutants (LRP16_D160A or LRP16_I161A) were treated with etoposide alone or in the presence of MRS2578. Their lysates were immunoblotted with the indicated antibody. One representative experiment of three was shown.

The online version of this article includes the following figure supplement(s) for figure 6:

**Figure supplement 1.** A small molecule MRS2578 inhibits the NF-κB activity induced by LRP16, and disrupts the interactions of the LRP16, PKR, and IKKβ complex in response to DNA damage.

dependently inhibited when cells were pretreated with MRS2578 (*Figure 6D* and *Figure 6—figure supplement 1D*). Taken together, these results suggested that CRC pretreated with MRS2578 induces cell killing and abrogates LRP16-mediated NF-κB activation in response to etoposide, leading to conversion of LRP16 from a survival into a killer molecule.

## Combination drug administration induces synergistic cytotoxicity ex vivo

To evaluate whether MRS2578 increased the sensitivity of CRC cells to etoposide, we first assessed the responses of CRC cells to a combination of etoposide and MRS2578 or each drug alone. At an unfixed ratio, the concentrations of MRS2578 and etoposide spanning the $IC_{50}$ of each cell line were selected for a combination study in which we calculated the combination index (CI) values with the ComboSyn software (CI <1, synergistic; CI = 1, additive; and CI >1, antagonism) (*Chou, 2010*). To more accurately analyze the degree of synergy between etoposide and MRS2578, a CI value was calculated. Of note, both SW480 and HCT116 cells treated with the combination of the two drugs showed synergistic cytotoxicity when assessed with the CalcuSyn model (*Figure 6E*). Cells stably overexpressing LRP16 displayed less sensitivity to etoposide alone than control cells, but these cells were even more sensitive to a combination of etoposide and MRS2578 (*Figure 6F–G*). To further assess whether the synergistic inhibition of cell proliferation by MRS 2578 and etoposide was modulated by enhanced apoptosis, we performed an Annexin-V/PI binding assay. Notably, the protection against apoptosis afforded by LRP16 overexpression in response to etoposide was offset, at least partially, when MRS2578 was used stimultaneously (*Figure 6H*). Additionally, results from cell-cycle profiling revealed that both cells stably expressing LRP16 and the control pretreated with MRS2578 displayed significant reductions in the proportions of cells in $G_0/G_1$ phase and showed significantly increased in the percentages of the cells in $G_2$ phase (*Figure 6I*). These findings reveal that MRS2578 increases the sensitivity of CRC cells to etoposide by enhancing their apoptosis and inhibiting cell growth.

To further investigate whether MRS2578 increased the sensitivity of CRC cells to IR, we selected two cell lines (DLD1 and HT-29) and exposed them to IR over time until resistance emerged, to generate the two cell lines DLD1_R (parental DLD1) and HT-29_R (parental HT-29). Compared with their parental cells, both the DLD1_R and HT-29_R cells were substantially sensitized to MRS2578 (*Figure 6J*), indicating that the cytostatic effect of MRS2578 (by inducing apoptosis and reducing cell growth) contributed to increasing the sensitivity or reducing the resistance of the cells to IR, and that the cytostatic effect was also independent of the IR resistance mechanism.

Our next step was to understand how MRS2578 blunts genotoxicity-induced NF-κB activation in CRC and significantly sensitizes tumor cells to DNA-damaging cytotoxic therapies. We have shown that LRP16, interacting with PKR, plays a role in extranuclear NF-κB signaling, but not in nuclear NF-

κB signaling, during the DNA damage response. Using LRP16 mutants (D160A and I161A), which function in a dominant inhibitory fashion by impeding the PAR-binding ability of LRP16, significantly reduced etoposide-induced NF-κB activity (*Figure 3F*). Thus, we evaluated the effect of MRS2578 on genotoxicity-induced NF-κB signaling and it is possible that this inhibitor behaves in similar fashion by impeding the PAR-binding ability of LRP16 to impede NF-κB activity. Of note, cells pretreated with MRS2578 significantly impeded the phosphorylation of IKKα and IKKβ induced by etoposide, similar to the phenomenon detected in cells stably expressing D160A and I161A (*Figure 6D*). Pretreating cells with MRS2578 profoundly reduced both the etoposide-stimulated phosphorylation of IKKα and IKKβ and the etoposide-stimulated autophosphorylation of PKR (*Figure 6K*). Furthermore, it also inhibited the etoposide-stimulated interaction among LRP16, PKR, and IKKβ and thus prevented the formation of a ternary complex, which might eventually impede the PKR activation of IKK (*Figure 6—figure supplement 1E*). Taken together, these results suggested that inhibiting the binding of LRP16 to PAR or pretreating cells with MRS 2578 is as effective as inhibiting PKR activity in blocking the biochemical effects of PKR in the cytoplasm and in inhibiting the NF-κB activity induced by LRP16 in response to DNA damage.

## Blockade of LRP16/NF-κB signaling inhibits the tumorigenicity of CRC cells in vivo

In light of our ex vivo findings, we examined the effect of LRP16 knockdown in vivo. Stable knockdown of LRP16 introduced by its shRNAs in SW620 cells suppressed tumor growth in the subcutaneous xenograft model. The mean weight of LRP16-silenced tumors was also reduced in SW620 cells compared with controls (*Figure 7A* and *Figure 7—figure supplement 1A–B*). Of note, compared with controls, stable knockdown of LRP16 was not only associated with a significant reduction in the growth of the primary SW620 tumors, but also with a marked increase in the sensitivity of the xenograft tumors to etoposide treatment (*Figure 7A* and *Figure 7—figure supplement 1B*). Silencing of LRP16 at mRNA levels in SW620 were verified by qPCR (*Figure 7—figure supplement 1C*). Body weight measurements made during the study indicated that etoposide treatment was tolerated by the animals (*Figure 7—figure supplement 1D*). Knockdown of LRP16 significantly inhibited cell proliferation and sensitized tumor cells to etoposide in SW620 cell xenograft models, as determined by Ki-67 staining. SW620 xenografts stably expressing shLRP16 showed the induction of apoptosis following etoposide treatment, as evidenced by the increased expression of terminal deoxynucleotidyl transferase dUTP nick-end labeling (TUNEL) (*Figure 7B*). Similar results were also obtained after IR treatment of the mouse xenograft tumors (*Figure 7—figure supplement 2A–E*), indicating that the expression of LRP16 is not only required, but also sufficient to promote radio-resistance in a CRC tumor model. Collectively, knockdown of LRP16 is synthetically lethal in CRC in vivo through suppressing cell growth and sensitizing the tumor to genotoxicity therapies.

## MRS2578 suppresses CRC growth in mouse xenograft models

Based on our findings obtained in cell culture models ex vivo, we next evaluated the anti-tumor effects of MRS2578 in mouse xenograft models in vivo. We first analyzed the toxicity of MRS2578 in mice. To define the appropriate doses for the in vivo experiments, mice were administered intraperitoneal injections of MRS2578 at tolerable doses (10 mg/kg or 20 mg/kg), as previously described (*Syhr et al., 2014*). At these doses, NF-κB activity could be inhibited in our cell culture models (*Figure 6D* and *Figure 6—figure supplement 1D*). The dose range between 10 and 20 mg/kg was tolerable, with no deaths recorded for up to 12 days of twice weekly administration. To test the potency of MRS2578 in vivo, CRC xenografts derived from SW620 cells were treated with increasing doses (0, 10, and 20 mg/kg, twice weekly) of MRS2578 via the intraperitoneal route for 28 days. The results showed that treatment of mice with MRS2578 resulted in a dose-dependent repression of CRC in vivo (*Figure 7Ca–Cb*). Hematoxylin and eosin (H and E) staining of the tumor tissues showed that the number of cells was substantially reduced in the tumors treated with MRS2578, and that the tumors were filled with fibrosis-like tissue. Consistent with this finding, an IHC analysis indicated that the expression of Ki67 was profoundly reduced in the tumor cells treated with MRS2578 (*Figure 7Cc*). No significant weight loss or organ toxicities were observed in the mice treated with MRS2578 (*Figure 7—figure supplement 3Aa–Ab*). These findings suggest that doses between 10

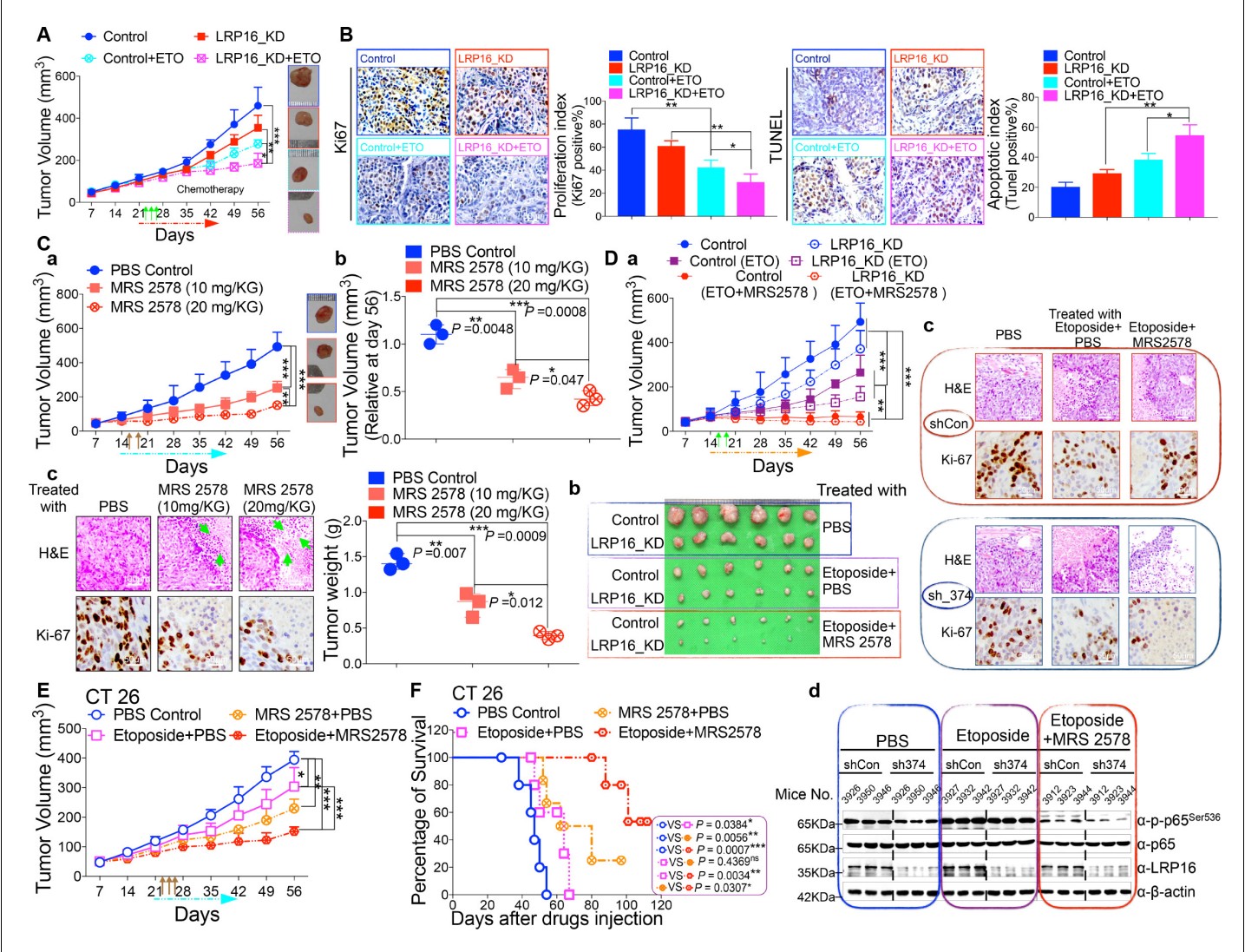

**Figure 7.** LRP16 deficiency sensitizes CRC to DNA-damaging cytotoxic therapies and combination of MRS2578 and etoposide synergistically represses CRC in xenograft models. (**A**) *nu/nu* nude mice were transplanted with 1 × 10⁶ SW620 cells infected with a lentivirus carrying either control shRNA or LRP16 shRNA. When the tumors were established, mice with tumor sizes > 150 mm³ (n = 5 per group) were treated with the control or etoposide (20 mg/kg intraperitoneal injection) for the indicated numbers of days. The sizes of the tumors were measured every 3 days with a Vernier caliper. (**B**) Representative immunohistochemical (IHC) staining for TUNEL and Ki67 in tumor tissues under the indicated conditions. The bar denotes 50 μm. Error bars in (**A**) and (**B**) represented the mean ± SD for triplicate experiments. *p<0.05, **p<0.01, ***p<0.001. (**Ca–Cc**) *nu/nu* nude mice with CRC cell xenografts were treated with increasing doses of MRS2578 (0, 10, or 20 mg/kg) twice a week for 28 days. Tumor volumes and bodyweights were measured weekly. At the end of the study, the mice were killed, and the tumors were removed and analyzed. All p values were compared with the control group. Error bars indicated the mean ± SD for triplicate experiments. *p<0.05, **p<0.01, ***p<0.001. (**D**) *nu/nu* nude mice were transplanted with CRC cells infected with a lentivirus carrying either control shRNA or LRP16 shRNA. When the tumors were established, the mice with tumor sizes > 150 mm³ (n = 6 per group) were treated with the control or etoposide (20 mg/kg intraperitoneal injection) and/or MRS2578 (10 mg/kg intraperitoneal injection) for the indicated number of days. Tumor volumes (**Da**), representative bright-field images of the tumors (**Db**), Representative IHC staining for Ki-67 in tumor tissues from mice after treatment with MRS2578 or a combination of etoposide and MRS2578. Scale bar represents 50 μm (**Dc**), representative tumors were isolated at the end of the assay, and tissue lysates were immunoblotted with the indicated antibodies (**Dd**). (**E**) BALB/c mice were implanted with CT26 colorectal adenocarcinoma cells. When the tumors were established, the mice with tumor sizes > 150 mm³ (n = 6 per group) were treated with the control, MRS2578, etoposide, or a combination of etoposide and MRS2578, for the indicated number of days. Error bars in (**D**) and (**E**) represented the mean ± SD for triplicate experiments.*p<0.05, **p<0.01, ***p<0.001. (**F**) Kaplan–Meier survival curves for BALB/c mice implanted CT26 colorectal adenocarcinoma xenografts after treatment with the control, MRS2578, etoposide, or a combination of etoposide and MRS2578. *p<0.05, **p<0.01, ***p<0.001. One representative experiment of three was shown.

The online version of this article includes the following figure supplement(s) for figure 7:

**Figure supplement 1.** LRP16 deletion sensitizes CRC to etoposdie in xenograft models.

*Figure 7 continued on next page*

*Figure 7 continued*

**Figure supplement 2.** LRP16 deficiency sensitizes CRC to IR in vivo.

**Figure supplement 3.** Evaluation of the toxicity and antitumor efficacy of MRS2578 in vivo and combination of MRS2578 and etoposide has no significant toxicity in vivo.

and 20 mg/kg provide the optimal therapeutic index for MRS2578 for in vivo experimentation involving CRC xenografts.

## MRS2578 synergizes with etoposide in the suppression of CRC in vivo

As further evidence for the important translational implications of the present studies, in in vivo therapy models in immune-deficient mice, our combinations of etoposide and MRS2578 yielded potent antitumor responses. First, for SW620 cells xenografts, as expected, mice treated with etoposide alone showed a significant reduction in tumor burden versus those treated with vehicle alone. However, the combination (simultaneous drug administration) treatment yielded a further tumor burden reduction compared with vehicle or etoposide alone (*Figure 7Da–Db* and *Figure 7—figure supplement 3Ba–Bb*). The treatment was well tolerated, as no significant weight loss and no organ toxicities were observed in mice that received these treatments, including combined treatment, throughout the study (*Figure 7—figure supplement 3Ba–Bb*). Of note, the combination of etoposide and MRS2578 resulted in significantly greater growth inhibition, as determined by Ki-67 staining (*Figure 7Dc*). Lastly, we also checked the effects of the small molecule inhibitor MRS2578 on oncogenic NF-κB signaling in xenograft tumors. As expected, phospho-p65 levels, but not total forms, are clearly decreased in the compound MRS2578 treated groups (*Figure 7Dd*). These results are consistent with the observations in the CRC cell experiments ex vivo.

We also investigated the combined effects of etoposide and MRS2578 on CT26 colorectal adenocarcinomas (BALB/c origin) implanted into BALB/c mice. Both the therapeutic effect and safety profile were evaluated. As expected, treatment with the combination of etoposide and MRS2578 further reduced tumor growth, indicating that this combination was more active than either single agent alone in inhibiting CRC cell growth and significantly prolonged mouse survival (*Figure 7E–F* and *Figure 7—figure supplement 3C*). There were no significant reductions in the mouse bodyweights, indicating that the toxicity of this combination is controllable (*Figure 7—figure supplement 3C*).

Taken together, these findings show that pharmacologically targeting LRP16/NF-κB signaling prevents tumorigenesis and suggest that the combination of MRS2578 and etoposide offers therapeutic opportunities for CRC and may warrant an immediate clinical trial.

## Discussion

As our understanding of the molecular basis of cancer has improved, a number of dysregulated signaling pathways responsible for driving disease progression have been identified. Efforts have been made to exploit these pathways as targets for therapeutic intervention, with the expectation that drugs capable of modulating them would deliver previously unachievable efficacy. However, the genetic instability and hypermutation rates of cancer, coupled with the redundancy often built into biological systems, have undermined the importance of singular targets. In the presence of this confluence of factors, resistant mutations arise and compensatory signaling pathways become upregulated, limiting the utility of specific inhibitors (*Ribic et al., 2003*). These observations suggest that while the development of new targets is critical, priority should be given to those that have the potential to act in synergy with and increase the therapeutic index of established treatment modalities. LRP16, as a cofactor for multiple nuclear receptors, fits this description (*Han et al., 2011*). LRP16 initially attracted attention because of the foundamental roles it plays in driving tumor progression; however more recently, relationships between the cofactor and other disease pathways have emerged. The intersection of these two features has thus made LRP16 an ideal oncology target.

A key pathway linking DNA damage with apoptosis, senescence and DNA repair mechanisms involves activating the NF-κB complex (*Bernard et al., 2004*; *Hayden and Ghosh, 2008*;

*McCool and Miyamoto, 2012*; *Wan and Lenardo, 2010*). It is widely accepted that genotoxicity-induced NF-κB activation is initiated by both ATM and PARP1, which trigger the phosphorylation and PIASy-mediated SUMOylation of nuclear NEMO. Our previous and present studies favor two possible roles of LRP16 in activating NF-κB (*Figure 8*): first, LRP16 functions not only by providing the lesion specificity during the cellular response to DNA damage by its unique interactions with Ku70/Ku80, but also ensures the successful PARP1/PAR-dependent recruitment of both ATM and PIASy, together with NEMO. These results underscore the function of LRP16, a versatile protein that preferentially occurs in the nucleus, in the early nuclear signaling cascade following DNA damage; second, when LRP16 detects DNA lesions and PAR, it assembles the PKR and IKKβ complex, the key players in NF-κB activation, in the cytoplasm. This complex can be formed by a direct protein–PAR interaction, as well as by protein–protein interactions. The PAR chains bound to LRP16 act as an interaction platform, which is required for PKR-mediated IKK activation and then activation of the NF-κB pathway. Thus, LRP16 appears to have a dual function, which might at least partially explain how these DNA-damage-initiated nuclear events are linked to the activation of cytoplasmic IKK.

Optimizing radiotherapy and chemotherapy for the treatment of malignant neoplasms has relied on the iterative development and testing of models involving tumor growth dynamics, mutation rates and cell-killing kinetics. However, the most theoretically effective tumoricidal strategies must usually be tempered because of detrimental effects to the host. This reality has led to the development of regimens in which therapies are administered at intervals or cycles to avoid irreparable damage to vital host functions. However, the recovery and repopulation of tumor cells between treatment cycles is a major cause of treatment failure (*Kim and Tannock, 2005*). Interestingly, rates of tumor cell repopulation have been shown to accelerate in the intervals between successive courses of treatment and solid tumors commonly show initial responses followed by rapid regrowth and subsequent resistance to further chemotherapy. Understanding the resistance mechanisms involved may open new therapeutic opportunities. Extensive previous research has demonstrated that cancer cells develop resistance to platinum drugs in a variety of ways, including the following: first, pre-target resistance, for instance, through the reduced accumulation or increased extrusion of etoposide by transporters; second, on-target resistance, caused by DNA repair; third, post-target

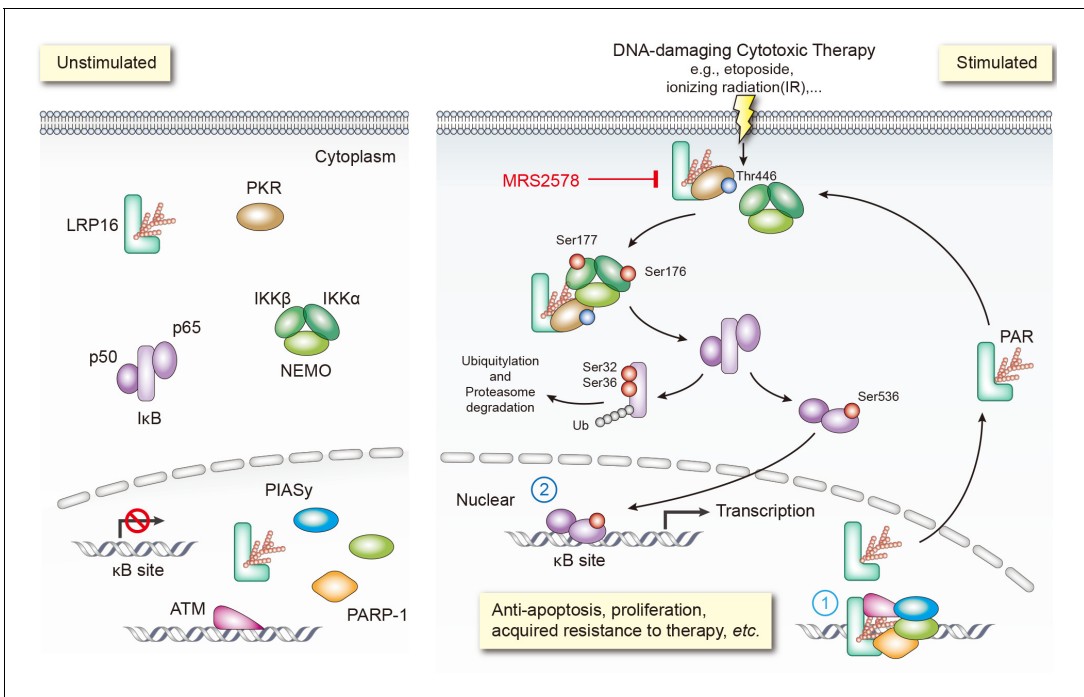

**Figure 8.** Schematic model of LRP16 function as a signaling molecule in the genotoxicity-initiated NF-κB signaling pathway. A model summarizing the role of the LRP16/PKR/NF-κB core signaling axis in reducing the sensitivity of colon cancer cells to DNA-damaging therapies, and the synergistic effects of MRS2578 and etoposide in suppressing CRC.

resistance, through the modulation of DNA damage recognition, the damage response, and apoptosis; and fourth, off-target resistance, for example, through compensatory pro-survival signals or non-specific adaptive responses that are not directly activated by drugs (*Martin et al., 2008*). The apoptotic capacities of chemotherapeutic agents have been widely used to determine the responses of cancer cells to them (*Johnstone et al., 2002*). Cellular apoptosis is a tightly regulated process that is controlled by numerous signal transduction pathways, including the NF-κB pathway (*Johnstone et al., 2002*). The NF-κB signaling pathway remains a very attractive target for pharmacological intervention because it has crucial functions in human health and disease, particularly in inflammatory diseases and cancers (*Hayden and Ghosh, 2008*; *Perkins, 2007*; *Smale, 2011*). Collectively, these studies support several conclusions: first, the outcomes of genotoxic exposures to any specific benign or neoplastic cell depend on the integration of innate damage response capabilities and the context that is dictated by the composition of the tumor microenvironment; second, although intrinsic drug resistance is clearly operative in some cancers, acquired resistance can also occur without alterations in intrinsic cellular chemosensitivity (*Davis and Tannock, 2000*; *Kim and Tannock, 2005*); and third, specific tumor microenvironment that promote therapy resistance are attractive targets for augmenting responses to more general genotoxic therapeutics. However, the complexity of the damage response program also supports strategies that are focused on inhibiting upstream master regulators, such as NF-κB, which may be more efficient and effective adjuncts to cytotoxic therapies, provided their side effects are tolerable.

Clinically, PARP1 inhibitors are a very exciting spectrum of drugs that are currently used in cancer management. Blocking the catalytic activity of PARP1 has been shown to inhibit base-excision repair (BER), resulting in the accumulation of SSBs, as well as DSBs, during DNA replication, and this damage, in turn, activates homologous recombination (HR) (*O'Connor, 2015*). Recent studies have shown that the disruption of any HR-related pathway, such as by *BRCA* mutations, and disruption of Fanconi anemia, and *ATM* genes, can predict the sensitivity of tumors to the inhibition of PARP1 by small-molecule inhibitors and the associated cytotoxicity (*D'Andrea, 2010*; *Mateo et al., 2015*; *O'Connor, 2015*; *Pommier et al., 2016*). Advances in understanding how and where, at a molecular level, these agents function optimally as cytotoxic agents and the recent progress in developing the best reagents are particularly important for the future use of PARP1 inhibitors in cancer therapy. The clinically available PARP1 inhibitors have shown considerable efficacy, especially in the treatment of breast and ovarian cancers, in patients with hereditary deletions of the HR *BRCA1/2* genes (*Bryant et al., 2005*; *Farmer et al., 2005*). Cancers presenting with these mutations represent 5–10% of all triple-negative breast cancers (estrogen, progesterone, and HER2 receptor-negative breast cancers; TNBCs) (*Bryant et al., 2005*; *Farmer et al., 2005*). However, the responses to PARP inhibitor therapy, even in *BRCA*-mutant breast cancers, have not been highly persistent. Furthermore, PARP inhibitors have failed to show impressive clinical benefits in patients with sporadic TNBCs and/or other cancers, suggesting that new strategies must be developed to maximize the efficacy of PARP inhibitors (*Lord and Ashworth, 2013*).

In summary, we discovered a model in which LRP16 selectively interacts and activates PKR and also acts as a scaffold to assist the formation of a ternary complex of PKR and IKKβ, prolonging the PAR-dependent NF-κB activation caused by genotoxic threats. Our preclinical data in CRC cell lines and mouse xenografts, as outlined in this study, suggest the potential for improving the clinical efficacies of DNA-damaging cytotoxic therapies by combining MRS2578 for patients with CRC. Our studies convincingly demonstrate that using small molecules to interfere with the binding of LRP16 to PKR and IKKβ, leading to the inactivation of its downstream NF-κB signaling pathways, is feasible with regard to the suppression of the proliferation of human CRC cancer cells. Thus, our results imply that LRP16 would not classically be viewed as 'undruggable', and open up an avenue that a small molecule targeting LRP16 would be a promising strategy to combat CRC in future therapies.

## Materials and methods

### Ethics sataement

Human Subjects: The human patient study was approved by the Ethics Committee of the Chinese PLA General Hospital (Beijing, China), and informed consent was obtained from all patients (S2016-

127-01). All samples were obtained in accordance with the Health Insurance Portability and Account-ability Act (HIPAA).

Animal experimentation: This study was performed in strict accordance with the recommendations in the Guide for the Care and Use of Laboratory Animals of the National Institutes of Health, and the protocol was approved by the Institutional Animal Care and Treatment Committee of the Chinese PLA General Hospital (IACTC-CPGH-062). All surgery were performed under sodium pentobarbital anesthesia, and every effort was made to minimize suffering.

## Clinical tissue specimens and immunohistochemistry (IHC) analysis

The samples of carcinomas and adjacent normal tissues were obtained from surgical specimens from patients with CRC for whom information on their clinicopathological characteristics were available and were approved by the department of pathology at the Chinese PLA General Hospital and the department of pathology at the Xiyuan Hospital of China Academy of Chinese Medical Sciences. The samples were frozen in liquid nitrogen immediately after their surgical removal until analysis. Colon tissue arrays were prepared and subjected to IHC analysis with the standard 3,3′-diaminobenzidine (DAB) staining protocol. All experiments were approved by the Ethics Committee of the Chinese PLA General Hospital (Beijing, China), and informed consent was obtained from all patients.

## Cell lines, antibodies, and reagents

All human CRC cell lines HCT116 (ATCC Cat# CCL-247, RRID:CVCL_0291), DLD1 (ATCC Cat# CCL-221, RRID:CVCL_0248), LoVo (ATCC Cat# CCL-229, RRID:CVCL_0399), LS180 (ATCC Cat# CL-187, RRID:CVCL_0397), RKO (ATCC Cat# CRL-2577, RRID:CVCL_0504), SW48 (ATCC Cat# CCL-231, RRID:CVCL_1724), CACO2 (ATCC Cat# HTB-37, RRID:CVCL_0025), HT-29 (ATCC Cat# HTB-38, RRID:CVCL_0320), SW1116 (ATCC Cat# CCL-233, RRID:CVCL_0544), SW480 (ATCC Cat# CCL-228, RRID:CVCL_0546), and SW620 (ATCC Cat# CCL-227, RRID:CVCL_0547) were obtained from the American Type Culture Collection (ATCC) (Manassas, VA) and the identities have been authenticated by short tandem repeat DNA profiling. All cells described above were regularly tested for mycoplasma contamination. Cells were cultured in RPMI 1640 medium containing 10% fetal calf serum, 2 M glutamine, 100 U/ml each of penicillin and streptomycin. The following antibodies were used in this study for immunoblotting and immunoprecipitation: antibodies directed against p-TAK1 (Thr184/187, Cat# 4508S, RRID:AB_561317) (1:1000), TAK1 (Cat# 5206S, RRID:AB_10694079) (1:1000), p-IKKα/IKKβ (Ser176/Ser177, Cat# 2078S, RRID:AB_2079379) (1:1000), IKKβ (Cat# 2370S, RRID:AB_2122154) (1:1000), NEMO (Cat# 2695S, RRID:AB_10695250) (1:1000), p-p65 (Ser536, Cat# 3033S, RRID:AB_331284) (1:1000), p53 (7F5, Cat# 2527S, RRID:AB_10695803) (1:1000), p-IκBα (Ser32/Ser36, Cat# 9246S, RRID:AB_2151442) (1:1000), PARP1 (46D11, Cat# 9532S, RRID:AB_10695538) (1:1000), Caspase 3 (8G10, Cat# 9665S, RRID:AB_2069872) (1:1000), PKR (D7F7, Cat# 12297, AB_2665515) (1:1000), and NF-κB1 p105/p50 (D4P4D, Cat# 13586, AB_2665516) (1:1000) were obtained from Cell Signaling Technology (Beverly, MA, USA); antibodies directed against p65 (Cat# sc-372, RRID:AB_632037) (1:1000), p50 (Cat# sc-114, RRID:AB_632034) (1:1000), p-JNK (Cat# sc-81502, RRID:AB_1127391) (1:1000), JNK (Cat# sc-7345, RRID:AB_675864) (1:1000), GST (Cat# sc-80998, RRID:AB_1124757), Sp1 (Cat# sc-17824, RRID:AB_628272) (1:2500), β-tubulin (Cat# sc-53140, RRID:AB_793543) (1:2000), β-actin (Cat# sc-69879, RRID:AB_1119529) (1:2000), and glyceraldehde-3-phosphate dehydrogenase GAPDH (Cat# sc-166545, RRID:AB_2107299) (1:3000) were obtained from Santa Cruz Biotechnology (Santa Cruz, CA, USA); antibody directed against p-PKR (Thr446) (Cat# 11280, AB_2665517) (1:1000) was obtained from Signaling Antibody (SAB, Baltimore, MD, USA); ant-FLAG antibody (Cat# SAB4200071, RRID:AB_10603396) was obtained from Sigma; Alexa-Fluor-488-conjugated goat anti-rabbit IgG antibody and Alexa-Fluor-594-conjugated goat anti-mouse IgG antibody were obtained from Life Technologies.

The following reagents were used in this study. Etoposide, 3-aminobenzamide (3-AB), PJ-34, benzamide (BEN), ethidium bromide (EB), and human recombinant TNF-α were purchased from Sigma-Aldrich (St. Louis, MO, USA). Doxorubicin and camptothecin were obtained from KeyGEN Biotech (Nanjing, China). Lipofectamine 3000 and Superfect Transfection Reagent were obtained from Invitrogen (Carlsbad, CA, USA) and Qiagen (Chatsworth, CA, USA), respectively. Protein A agarose was obtained from the Millipore Corporation (Bedford, MA, USA). PhosSTOP Inhibitor Cocktail Tablets and Complete Protease Inhibitor Cocktail Tablets were obtained from Roche Applied Science

(Mannheim, Germany). Biotin-NAD$^+$, NAD, human PARP enzyme, activated DNA, and PARP buffer were purchased from Trevigen (Gaithersburg, MD, USA). GST Bind Resin was obtained from Novagen (Madison, WI, USA). CCK-8 was obtained from Dojindo (Tokyo, Japan). Apoptosis Detection Kit was obtained from BD Bioscience (San Jose, CA, USA).

## Plasmids and transfection reagents

The constructs expressing wild-type LRP16 and LRP16 mutants have been described in our previous report (**Han et al., 2003**, **Han et al., 2007**; **Wu et al., 2015**; **Yang et al., 2009**). Wild-type PKR and its mutants were cloned by PCR amplification into pcDNA3–FLAG. siRNA-resistant LRP16 was cloned by PCR amplification into pcDNA3–FLAG. The dominant-negative mutant IκBm (IκBSR), as described previously (**Wu et al., 2015**), in which serines 32 and 36 were mutated to alanine, was inserted into the pcDNA3–FLAG vector. The identities of all the constructs used in this study were verified by DNA sequencing. Cells were transfected with the indicated plasmids using Superfect Reagent, as described previously (**Wu et al., 2011**, **Wu et al., 2015**). For siRNA transfection, cells were plated at 30–60% confluence in Opti-MEM serum-free medium and transfected with a specific siRNA duplex using Lipofectamine RNAiMAX Transfection Reagent (Life Technologies, Paisley, UK), according to the manufacturer's instructions for 48 hr. siRNAs were ordered as reverse phase (RP)-HPLC-purified duplexes from GenePharma (Shanghai, China). The sequences were as follows:

LRP16-374, 5′-GCAGCGGGGAGGAACAUUAC-3′,
LRP16-668, 5′-GACUGGCAAGGCCAAGAUC-3′,
siPKR-1, 5′-GCAGGGAGUAGUACUUAAA-3′,
siPKR-2, 5′-GCAUGGGCCAGAAGGAUUU −3′,
siPKR-3, 5′-GCAGAUACAUCAGAGAUAA −3′, and
siPKR-4,5′-CCUGAGACCAGUGAUGAUU −3′.

## Apoptosis assay and cell proliferation assay (Edu staining)

Cells were plated and treated for 48 hr with the indicated reagents; $1 \times 10^6$ cells were washed with cold PBS and resuspended in 100 μl of binding buffer. An apoptosis assay was performed with the Annexin-V and propidium iodide using the protocol provided with the apoptosis detection kit (BD Biosciences). Cells without Annexin V or propidium iodide were used to detect autofluorescence. EdU staining was performed using the Click-iT EdU Alexa Fluor 488 Imaging Kit (Invitrogen), according to the manufacturer's protocol. EdU was added directly to the culture medium at a final concentration of 10 μM and incubated for 2 hr. The cells were fixed with 4% paraformaldehyde in PBS for 15 min, and then permeabilized with 0.5% Triton X-100 for 20 min. Nuclei were counterstained with Hoechst 33342. Three biological replicates were prepared for each treatment and each assay was performed in triplicate.

## RNA isolation and quantitative real-time PCR (RT-qPCR) analysis

RNA from the cell lines was isolated with TRIzol Reagent and purified with the RNeasy Mini Kit according to the manufacturer's protocol (Qiagen). Aliquots of 1 μg of total RNA were reverse-transcribed with the ThermoScript RT–PCR System (Invitrogen). RT–qPCR was performed according to the instruction for SYBR Green PCR Master Mix (Applied Biosystems, Foster City, CA, USA) with the ViiA 7 Real-Time PCR System (Applied Biosystems). Relative expression levels were calculated using the $2^{-\Delta\Delta CT}$ method. β-Actin was used as the housekeeping gene for normalization. The results for these experiments were all reproducible and therefore only one of each experiment is presented.

## Cell viability assay and clonogenic assay

Cells were seeded in 96-well plates overnight and treated with the indicated drugs. A CCK8 assay was performed after incubation for 72 hr. Untreated cells were used to indicate 100% cell viability. The absorbance (optical density, OD) was read at a wavelength of 450 nm on an enzyme-linked immunosorbent assay (ELISA) plate reader. The cell viability rate was calculated as follows: (OD treated/OD control)×100%. The IC$_{50}$ values were calculated with GraphPad Prism version 6.0 (GraphPad Software, La Jolla, CA, USA). The cells were trypsinized to generate a single-cell suspension, and seeded in six-well plates at 104 cells per well, and treated with or without the indicated drugs. At 2–3 weeks after seeding, the colonies were stained with crystal violet as described

previously (*Li et al., 2013*). Three biological replicates were prepared for each treatment and each assay was performed in triplicate.

## Immunofluorescence microscopy

The cells were immunostained as described previously (*Wu et al., 2011*, *Wu et al., 2015*). Briefly, the cells were fixed in 4% paraformaldehyde (PFA) in PBS. After blocking and permeabilization with 0.3% Triton X-100% and 1% bovine serum albumin in PBS, the cells were probed with the indicated antibodies. Alexa-Fluor-488- and Alexa-Fluor-594-labeled secondary antibodies were used to visualize the immunofluorescent signals. Representative fluorescent images were acquired with a confocal laser scanning microscope (FV1000, Olympus, Tokyo, Japan).

## Western blot analysis and immunoprecipitation (IP) assays

Cells were washed with ice-cold PBS and lysed on ice for 30 min with RIPA lysis buffer supplemented with protease inhibitors and PhosSTOP inhibitors. Protein concentrations were determined with the Bradford Protein Assay Kit (Bio-Rad, Carlsbad, CA, USA) and a calibration standard curve created with BSA. The samples were prepared for loading by adding 4 × sample buffer (Invitrogen) and heating the samples at 90°C for 10 min. The total proteins were separated with SDS-PAGE. The proteins in the gel were electrophoretically transferred to a PVDF membrane (Millipore), and then the membrane was blocked in 5% milk or BSA with Tris-buffered saline/Triton X-100 buffer (100 mM Tris-HCl [pH 7.4], 500 mM NaCl, 0.1% Triton X-100) (TBS-T 0.1%). The membranes were incubated overnight at 4°C with primary antibodies in 5% milk or BSA in 0.1% TBS-T. Horseradish peroxidase (HRP)-conjugated secondary antibody was added and incubated for 1 hr at room temperature in 0.1% TBS-T, and the signals were visualized with an enhanced chemiluminescence system. Immunoprecipitation (IP) experiments were performed, essentially as described previously (*Wu et al., 2011*, *Wu et al., 2015*; *Yang et al., 2009*). Briefly, for conventional IP experiments, cells were cultured to 70% confluence in 10 cm culture dishes. The cells were then either harvested without additional treatments or after treatment with either IR or a specific genotoxic agent. The cells were then lysed with IP buffer (50 mM Tris-HCl [pH 7.4], with 150 mM NaCl, 1 mM EDTA, and 1% Triton X-100) in the presence of a protease inhibitor mixture. After a preclearing step with protein A/G-agarose beads (Upstate), the protein lysate (1 mg) was immunoprecipitated with the agarose-immobilized antibody overnight at 4°C. The blot was incubated with HRP-conjugated secondary antibody, and the signals were visualized with an enhanced chemiluminescence system as described by the manufacturer. Similar results were obtained from two independent sets of experiments, and the result of one experiment is presented.

## Luciferase assay

Cells were seeded in 12-well plates. Experiments were set up as described previously (*Wu et al., 2011*, *Wu et al., 2015*). Briefly, the cells were then transfected with a luciferase reporter plasmid containing 3 × κB–luc motifs and the indicated plasmids and/or siRNAs together with pRL-SV40. The total amount of input DNA for each treatment was kept constant by supplementing it with pcDNA3.1. At 48 hr after transfection, the cells were either left untreated or treated with genotoxic stress. At the time of harvest, the promoter activity was assessed with a dual-luciferase assay kit. Briefly, the feeding medium was removed from the wells, and the cells were washed once with ice cold PBS and lysed with 100 μl of ice-cold reporter lysis buffer. The cell lysate (10 μl) was then added to 50 μl of luciferase substrate solution, after which 50 μl of Stop and Glow Buffer was added to visualize the luciferase signal. The bioluminescence generated was measured with a luminometer (Berthold Detection System, Pforzheim, Germany). The luminescence readings obtained were normalized to the protein concentration of the corresponding cell lysate and presented as the fold difference relative to the control setup.

## Recombinant protein expression and purification

For bacterial expression, IκBα cDNA (encoding aa 1–66) was cloned into pGEX-6p-1. The recombinant fusion protein expressed by the pGEX-6p-1–LRP16 plasmid has been described in our previous report (*Wu et al., 2011*, *Wu et al., 2015*). The procedure used to obtain the purified recombinant proteins from *Escherichia coli* cells has been described previously (*Wu et al., 2015*). *Escherichia coli*

BL21 cells were transformed with pGEX-6p-1, pGEX-6p-1-LRP16, or pGEX-6p-1-IκBα (aa 1–66) plasmid, and at $OD_{600}$ = 0.5–1.0, the cultures were induced with 100 mM IPTG for 10 hr at 20℃. The crude bacterial lysates were prepared by sonication (nine rounds of 20 s bursts on ice) in lysis buffer (40 mM Tris-HCl [pH 7.5], 150 mM NaCl, 1 mM EDTA, 0.5% NP-40, and 10% [v/v] glycerol) in the presence of a protease inhibitor mixture. The GST-tagged recombinant proteins were then purified using GST Bind Resin according to the manufacturer's protocol.

## GST pull-down assay

The GST-tagged proteins that were associated with the GST Bind Resin were incubated with various proteins of interest. The GST pull-down assay was performed as described previously (*Wu et al., 2011*, *Wu et al., 2015*). The total proteins that were associated with the beads were then analyzed by Western blotting analysis using the appropriate antibodies. Each GST pull-down sample was analyzed three times and reproducible results were obtained.

## Mass spectrometry

The bead-associated FLAG-tagged proteins of interest were eluted by incubation with the 3 × FLAG Peptide antigen (Sigma). The eluted products were fractionated by an SDS-PAGE. The gel was then subjected to silver staining, and the visible bands were excised and subjected to a mass-spectrometric analysis.

## Synthesis of polymers of ADP-Ribose (PAR) and PAR-binding assay

A 100 μl reaction mixture containing 3 μl of human PARP1 enzyme, 3 μl of 20 mM NAD or 250 μM biotin–NAD, 10 μl of 10 × activated DNA, and 5 μl of 20 × PARP buffer was incubated at 25℃ for 1 hr. The reaction mixture was then used directly in a PAR-binding assay. The recombinant proteins were incubated for 30 min at 32℃ in 40 mM Tris-HCl (pH 7.5), 150 mM NaCl, 1 mM EDTA, 0.5% NP40, 10% (v/v) glycerol, biotin-labeled PAR and/or small molecules. The reaction mixtures were applied to nitrocellulose and washed for 30 min with TBS-T containing 100 nM NaCl. After incubation with streptavidin–HRP, the bound biotin-labeled PAR was detected with a DAB Horseradish Peroxidase Color Development Kit. Each sample was detected three times. Each experiment was performed three times, and reproducible results were obtained.

## Combination therapy and criteria for synergism

Cells were plated on 96-well plates and treated with various concentrations of each drug alone or in combination, at a constant ratio. Following daily treatments for 4 days, the cells were assayed with CellTiter 96 Aqueous One Solution Cell Proliferation Reagent (Promega, Madison, Wi, USA). The absorbance values were used to determine the fraction of cells affected by each treatment and to determine the combination indices (CIs) according to the Chou–Talalay method, using CompuSyn software: CI <1 indicates synergism, CI = 1 indicates an additive effect, and CI >1 indicates an antagonistic effect.

## Gene expression microarray data analysis

The total RNA from colon cancer cells stably transfected with the control vector or LRP16-expressing plasmid, and treated with or without etoposide (50 μM) for the indicated periods, was isolated and purified with an RNeasy Kit (Qiagen, Hilden, Germany). The integrity of the RNA was assessed with an Agilent BioAnalyzer 2100 (Agilent Technologies, Palo Alto, CA, USA). The samples were processed and hybridized to an Affymetrix GeneChip Human Transcriptome Array (HTA) 2.0, which contains >6.0 million probes covering the exons of >65,000 coding and noncoding transcripts. The gene expression microarray data were preprocessed and normalized with the Affymetrix Expression Console software (V1.3.1) using the RMA sketch method. Several sources of gene sets were selected, including gene ontology (GO) and Kyoto Encyclopedia of Genes and Genomes (KEGG). The sets included groups based on molecular function, cellular localization, biological processes, and signaling pathways. The resulting lists were examined for their enrichment in terms of the GO (biological process) and KEGG pathways. For the latter, pathways associated with diseases were filtered out as reported (*Li et al., 2013*, *Li et al., 2014*). The enrichment analysis was based on a

hypergeometric test. p-Values were adjusted using Benjamini–Hochberg's false discovery rate (FDR); only FDRs <0.1 were considered. A correction for genes in overlapping clusters was applied.

## Hematoxylin and eosin (H and E) staining and IHC analysis

Tumor tissues were fixed in 4% formaldehyde solution and processed routinely for paraffin embedding. Sections were cut to a thickness of approximately 4 µm, placed on glass slides, and counterstained with H and E. For IHC, formalin-fixed paraffin-embedded tissue sections were dewaxed, hydrated, heated for 2 min in a conventional pressure cooker, treated with 3% $H_2O_2$ for 20 min, and then incubated with normal goat serum for 30 min. The sections were then incubated with antibodies overnight. After washing, the sections were incubated with biotin-labeled secondary antibody for 20 min at 37°C. The slides were then rinsed and incubated with streptavidin–biotin–peroxidase for 20 min. The color reaction was developed with 3,3′-diaminobenzidine tetrahydrochloride. The IHC specimens were independently assessed by two pathologists who were blinded to the origin of the samples, including the clinicopathological data. The staining intensity and extent of the stained area were graded according to the German semiquantitative scoring system: staining intensity of the nucleus, cytoplasm, and/or membrane (no staining = 0; weak staining = 1; moderate staining = 2; strong staining = 3); the extent of stained cells (0% = 0, 1%–24% = 1, 25%–49% = 2, 50%–74% = 3, 75%–100% = 4). The final immunoreactive score (0 to 12) was determined by multiplying the intensity score by the extent of stained cells. Based on the QS, LRP16 expression was graded as low (0–1) or high (2–3).

## In vitro kinase assay

SW620 cells were transfected with either pooled LRP16 siRNAs or pooled PKR siRNAs for knockdown. After 36 hr, the cells were lysed with IP buffer, and the kinase complex was prepared with an anti-NEMO antibody. GST–IκBα (aa 1–66) was purified on a glutathione–agarose column and used as the substrate. The reaction was performed with a mixture of the kinase complex, 0.5 µg of GST–IκBα (aa 1–66), and 100 µM ATP in kinase buffer (25 mM Tris [pH 7.5], 10 mM MgCl2, 2 mM EGTA, 1 mM dithiothreitol, and 1 mM sodium orthovandate) + inhibitors (0.5 mM PMSF, 10 mM β-glycerol phosphate (BGP), 300 µM sodium othovanadate, 1 µg/ml leupeptin, 1 µg/ml aprotinin, 10 mM sodium fluoride, 10 mM p-nitrophenyl phosphate) at 30°C for 30 min. The reactions were stopped by the addition of 10 mM EDTA and stored at −20°C until analysis by immunoblotting. The purified IKK complex and GST–IκBα (aa 1–66) were analyzed by Western blotting with the appropriate antibodies. Each sample was detected two times. Each experiment was performed two times and reproducible results were obtained.

## Animal models

Tumor cell xenografts were generated as previously described (*Li et al., 2013*). Female 6–8 week-old athymic nu/nu nude mice (RRID:IMSR_CRL:088) and BALB/c mice (RRID:IMSR_CRL:28) were purchased from HFK Bioscience Co., Ltd (Beijing, China). Mice CT26 (ATCC Cat# CRL-2638, RRID: CVCL_7256) was also obtained from ATCC (Manassas, VA) and the identities have been authenticated by short tandem repeat DNA profiling. Cells described above were regularly tested for mycoplasma contamination. All studies were approved by the Institutional Animal Care and Treatment Committee of the Chinese PLA General Hospital.

## Xenografts

Tumor cells ($3 \times 10^6$) were injected subcutaneously into the lateral flanks of mice and allowed to develop for 10–15 days. The tumor-bearing mice were randomized into groups and began treatment when the average tumor volume reached 100–150 $mm^3$. The mice with colon cancer xenografts were treated with etoposide, MRS 2578, or their combination, given intraperitoneally, at the indicated dose or with IR. For the combination treatment, both drugs were given concurrently. During treatment, the tumor volume (V) was measured with a Vernier caliper 2–3 times per week and calculated with the formula $V = (L \times W^2)/2$, where L is the length and W is the width, as previously described (*Li et al., 2013*). The volumes of the tumors treated with the vehicle control or the compounds were compared, and the p values were determined with a two-tailed Student's *t* test. At the

completion of the study, the mice were killed and necropsied, and the tumor tissues were removed for further analysis.

## Statistical analysis

All statistical analyses were performed with GraphPad Prism version 6.0 (GraphPad Software, La Jolla, CA, USA). Two-tailed Student's $t$ test, two-way ANOVA, or one-way ANOVA with Dunnett's multiple comparisons test was used to compare the statistical differences between the relevant groups. The synergistic effect in the animal study was analyzed with two-way ANOVA. A p value < 0.05 was considered statistically significant. The half maximal inhibitory concentration ($IC_{50}$) values were calculated with GraphPad Prism version 6.0. For all in vivo experiments, we used the GraphPad Prism software (ANOVA/Mann–Whitney test) to calculate the statistical significance of differences.

## Accession numbers

Microarray data were submitted to the NCBI Gene Expression Omnibus under accession number GSE93625.

## Acknowledgements

We thank prof. Chen Wang (Institute of Biochemistry and Cell Biology, Chinese Academy of Science, Shanghai) for providing the κB–luc plasmid. We also thank prof. Yufang Shi (Institute of Translational Medicine, Soochow University, Suzhou) and prof. Mingzhou Guo (Department of Gastroenterology and Hepatology, Chinese PLA General Hospital) for critical reading and appraisal of the manuscript. This work was supported by the grants from the National Natural Science Foundation of China (Nos. 81230061 to WDH, 81672797 to XLL, 81472612 to ZQW) and the Beijing Nova Program (No. Z141107001814104 to XLL) and the Beijing Nova Program Interdisciplinary Studies Cooperative Project (No. Z161100004916043 to XLL) and the National Key Research and Development Program of China (Nos. 2016YFC1303501 and 2016YFC1303504 to WDH).

## Additional information

### Funding

| Funder | Grant reference number | Author |
| --- | --- | --- |
| National Natural Science Foundation of China | 81230061 | Weidong Han |
| Beijing Nova Program | Z141107001814104 | Xiaolei Li |
| National Natural Science Foundation of China | 81672797 | Xiaolei Li |
| National Natural Science Foundation of China | 81472612 | Zhiqiang Wu |
| National Key Research and Development Program of China | 2016YFC1303501 and 2016YFC1303504 | Weidong Han |
| Beijing Nova Program Interdisciplinary Studies Cooperative Project | Z161100004916043 | Xiaolei Li |

The funders had no role in study design, data collection and interpretation, or the decision to submit the work for publication.

### Author contributions

Xiaolei Li, Conceptualization, Supervision, Funding acquisition, Project administration, Writing—review and editing; Zhiqiang Wu, Data curation, Funding acquisition, Investigation, Methodology, Writing—original draft, Project administration; Xiaojing An, Data curation, Formal analysis, Funding acquisition, Validation, Investigation, Methodology, Project administration; Qian Mei, Resources,

Data curation, Validation, Methodology; Miaomiao Bai, Data curation, Formal analysis, Investigation, Methodology; Leena Hanski, Data curation, Methodology; Xiang Li, Investigation, Methodology, helped with screening the small molecules; Tero Ahola, Investigation, Methodology; Weidong Han, Conceptualization, Data curation, Supervision, helped with screening the small molecules

### Author ORCIDs
Weidong Han http://orcid.org/0000-0003-3207-3899

### Ethics
Human subjects: Human Subjects: The human patient study was approved by the Ethics Committee of the Chinese PLA General Hospital (Beijing, China), and informed consent was obtained from all patients (S2016-127-01). All samples were obtained in accordance with the Health Insurance Portability and Accountability Act (HIPAA).

Animal experimentation: Animal experimentation: This study was performed in strict accordance with the recommendations in the Guide for the Care and Use of Laboratory Animals of the National Institutes of Health, and the protocol was approved by the Institutional Animal Care and Treatment Committee of the Chinese PLA General Hospital (IACTC-CPGH-062). All surgery was performed under sodium pentobarbital anesthesia, and every effort was made to minimize suffering.

### Decision letter and Author response
Decision letter https://doi.org/10.7554/eLife.27301.sa1
Author response https://doi.org/10.7554/eLife.27301.sa2

## Additional files
### Data availability
The following dataset was generated:

| Author(s) | Year | Dataset title | Dataset URL | Database and Identifier |
|---|---|---|---|---|
| Xiaolei Li, Zhiqiang Wu, Weidong Han | 2017 | Human colon cancer SW480 cells: stably transfected with vector control and LRP16 treated with or without etoposide | https://www.ncbi.nlm.nih.gov/geo/query/acc.cgi?acc=GSE93625 | NCBI Gene Expression Omnibus, GSE93625 |

The following previously published dataset was used:

| Author(s) | Year | Dataset title | Dataset URL | Database and Identifier |
|---|---|---|---|---|
| Muzny DM, Bainbridge MN, Chang K, Dinh HH, Drummond JA, et al | 2012 | Comprehensive Molecular Characterization of Human Colon and Rectal Cancer | https://tcga-data.nci.nih.gov/docs/publications/coadread_2012/ | TCGA Data Portal, coadread_2012 |

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
