## [Decision Letter]

Thank you for submitting your article "Blockade of the LRP16-PKR-NF-κB signaling axis sensitizes tumor cells to DNA-damaging cytotoxic therapy" for consideration by *eLife*. Your article has been reviewed by three peer reviewers, and the evaluation has been overseen by a Reviewing Editor and Kevin Struhl as the Senior Editor. The following individual involved in review of your submission has agreed to reveal his identity: Qinong Ye (Reviewer #3).

The reviewers have discussed the reviews with one another and the Reviewing Editor has drafted this decision to help you prepare a revised submission.

Summary:

The reviewers find that the paper describes the oncogenic role of LRP16 in prolonging the PAR-dependent NF-κB activation, thus proposing LRP16 as a potential therapeutic target in colorectal cancer (CRC). The authors identified a small molecule MRS2578 to strikingly abrogate the binding of LRP16 to PKR and IKKβ, thus inhibiting the oncogenic role of LRP16. The authors demonstrated that the combination of etoposide and MRS2578 (LRP16 inhibition) exhibits synergistic tumor cytotoxicity in CRC xenograft models. These data provide a new therapeutic strategy that sensitizes CRC cells to DNA damaging agent via targeting LRP16.

Essential revisions:

The reviewers raise a number of concerns that must be adequately addressed before the paper can be accepted.

1) The authors should increase the number of cases in the cohort of CRC patients to produce more convincing numbers.

2) The authors need to address whether PARP inhibitors (e.g. PJ-34 or 3-AB) for PAR inhibition cloud blunt the protective effects of LRP16 on CRC cell viability.

3) The biological function of LRP16 mutants (D160A or I161A) on CRC cells should be performed and then further confirm NF-κB activation mediated by LRP16 plays crucial roles in tumor cells resistance to DNA damage therapies.

4) Determination of LRP16 levels, p65 phosphorylation and XIAP in some representative colorectal carcinoma tissues and adjacent normal colon tissues by Western blots are needed to further support their conclusions. Alternatively, the specificity of the antibodies used may be confirmed.

5) The authors should further analyze whether LRP16 might similarly regulate NF-κB pathway at the transcriptional level under the same condition (e.g. κb-Luc activity).

6) The authors investigated the toxicity of MRS2578 or a combination of MRS2578 and etoposide in mice. However, these conclusions were only obtained from liver or kidney tissues in mice models. Whether other organs (e.g. heart, spleen, colon, etc.) display some toxicity should be confirmed.

*Reviewer #1:*

In this study, Li et al. demonstrated the oncogenic role of LRP16 and that it is a potential therapeutic target in colorectal cancer (CRC). They provide compelling evidences that LRP16 is a key factor to activate NF-κB signaling induced by etoposide. They also identified a small molecule to inhibit LRP16, and showed that the combination of etoposide and LRP16 inhibition synergistically suppressed CRC cell growth in vitro and in vivo. Although etoposide is not currently used to treat CRC, their data provide a new therapeutic strategy that sensitizes CRC cells to DNA damaging agent via targeting LRP16. Overall, this study is well designed, experiments were carefully performed, and results a quite solid to support their conclusion.

*Reviewer #2:*

In this manuscript Li et al. report that LRP16 is a key NF-κB regulator in genotoxicity-initiated signaling pathway and confers tumor cells acquired resistance to DNA damage therapies. Mechanistically, LRP16 interacts with and activates PKR, and then regulates the phosphorylated forms, but not the total forms, of the upstream regulators of the NF-κB pathway, IKKα and IKKβ, and also increased the levels of phospho-p65, leading to active NF-κB following DNA damage response. Finally, the authors report a small molecule, MRS2578 strikingly abrogated the binding of LRP16 to PKR and IKKβ, converting LRP16 into a death molecule and forestalling tumorigenesis. Inclusion of MRS2578 with etoposide, versus each drug alone, exhibits synergistic tumor cytotoxicity in CRC xenograft models.

In general, this is a very interesting study whose conclusions are supported by the presented results. However, as described below, there are several additional experiments and analyses that should be performed to strengthen the main conclusions.

1) The initial data on the cohort of CRC patients are limited and the significance of the correlation is somewhat borderline. So the authors should increase the number of cases to produce more convincing numbers.

2) The authors claim that LRP16 could prolong the polymers of ADP-ribose (PAR)-dependent NF-κB transactivation so that confers tumor cells acquired resistance to etoposide. Thus, the authors need to address whether PARP inhibitors (e.g. PJ-34 or 3-AB) for PAR inhibition cloud blunt the protective effects of LRP16 on CRC cell viability.

3) Although the previous (Wu et al., 2015) and present studies from the authors' own lab have demonstrated that LRP16 wide type, but not mutants (D160A or I161A), which were sufficient to significantly reduce its affinity to PAR, enhances NF-κB activation after etoposide treatment, however, the biological function of LRP16 mutants (D160A or I161A) on CRC cells should be performed and then further confirm NF-κB activation mediated by LRP16 plays crucial roles in tumor cells resistance to DNA damage therapies.

4) In Figure 6G, the authors generate the two cell lines DLD1_R (parental DLD1) and HT-29_R (parental HT-29) resistance to IR, which marker does the authors use to determine the occurrence of resistance to IR?

5) The difference between a small molecule MRS2578 selected in this submitted manuscript and traditional PARP inhibitors should be discussed in the Discussion section.

6) There are some spelling mistakes in this submitted manuscript, the authors should check them carefully.

7) The figures labels cited in this submitted manuscript are not consistent, the authors should thoroughly reviewed and modified according to the format of *eLife* journal, for example, "Supplementary Figure S7D"should be "Figure 7—figure supplement 1D".

8) Additionally, the supplement figures embedded in one picture are easily confused for readers. In order to increase readability, some supplement figures could be divided into several graphs according to the format of *eLife* journal, such as, Figure 1—figure supplement 1, Figure 1—figure supplement 2, Figure 1—figure supplement 3.

*Reviewer #3:*

The authors proposed a model in which LRP16 regulates NF-κB signaling in response to DNA damage. LRP16 selectively interacts and activates PKR, and acts as a scaffold to facilitate the ternary complex formation of PKR and IKK, prolonging the PAR-dependent NF-κB activation caused by DNA damage agents. LRP16 knockdown cells are hypersensitive to genotoxicity in vitro and in vivo. Moreover, the authors identified a small molecule, MRS2578, which strikingly abrogated the binding of LRP16 to PKR and IKK, thus blocking tumor growth. A combined use of MRS2578 and etoposide exhibited synergistic anti-tumor activity. The findings are interesting and the manuscript is generally well organized. The data provided support support their conclusions.

1) Using immunohistochemical staining, the authors claimed that the expression of LRP16 was significantly higher in the colon carcinoma samples than in the adjacent normal colon tissues. The elevated levels of LRP16 were positively correlated with NF-κB activation (positive p65 nuclear staining represents constitutive activation of NF-κB), but not with XIAP. To make these data more convincing, determination of LRP16 levels, p65 phosphorylation and XIAP in some representative colorectal carcinoma tissues and adjacent normal colon tissues by Western blots are needed to further support their conclusions. Alternatively, the specificity of the antibodies used may be confirmed.

2) Most of the conclusions are based on Western blots. Reintroduction of LRP16 significantly increased the phosphorylated forms, but not the total forms, of the upstream regulators of the NF-κB pathway, IKKα and IKKβ, and also increased the levels of phospho-p65. All changes focused on the upstream regulators of the NF-κB pathway. The authors should further analyze whether LRP16 might similarly regulate NF-κB pathway at the transcriptional level under the same condition (e.g. κb-Luc activity).

3) The authors identified a small molecule that could inhibit LRP16-mediated NF-κB activation upon treatment of DNA damage agents by immunoblot. Further analysis, such as κb-Luc activity, is required to show the effect of the small molecule.

4) The authors demonstrated that a small molecule targeting LRP16/NF-κB signaling prevented tumor growth and suggested that the combination of MRS2578 and etoposide offers therapeutic opportunities for CRC. In the first experiment, the authors investigated the toxicity of MRS2578 or a combination of MRS2578 and etoposide in mice. However, these conclusions were only obtained from liver or kidney tissues in mice models. Whether other organs (e.g. heart, spleen, colon, etc.) display some toxicity should be confirmed.

---

## [Author Response]

Essential revisions:The reviewers raise a number of concerns that must be adequately addressed before the paper can be accepted.1) The authors should increase the number of cases in the cohort of CRC patients to produce more convincing numbers.

We appreciate the reviewers for pointing out increasing the number of cases in the cohort of CRC patients makes our data more convincing. As suggested by the reviewers, we have already reviewed another cohort of patients with CRC containing 100 clinical tumor specimens with paired adjacent normal colon tissues from patients with CRC. Thus, all data from CRC clinical specimens were obtained from 202 clinical tumor specimens with paired adjacent normal colon tissues from patients with CRC. We have also analyzed the levels of LRP16, p65, XIAP and PKR expression in these 100 CRC clinical specimens, and all data obtained from these clinical specimens have been integrated into our previous results. Indeed, our presented results, as well as data obtained from additional experiments, demonstrated that LRP16 is overexpressed in human CRC samples compared with adjacent normal samples, and also confirms the critical role of LRP16 in promoting CRC tumorigenesis. These results have been included in the revised manuscript.

2) The authors need to address whether PARP inhibitors (e.g. PJ-34 or 3-AB) for PAR inhibition cloud blunt the protective effects of LRP16 on CRC cell viability.

We appreciate the point made by the reviewers. The reviewer is asking an excellent question. We will test whether PARP inhibitors (e.g. PJ-34 or 3-AB) for PAR inhibition could blunt the protective effects of LRP16 on CRC cell viability. Our presented results showed that LRP16 plays a critical role in DNA damage-initiated and PAR-dependent NF-κB transactivation. We thus performed additional cell viability and clonogenicity experiments to further support the protective effects of LRP16 on CRC cell viability in a PAR-dependent manner. Consistent with results from LRP16 mutants (D160A or I161A), PARP inhibition by PJ-34 or 3-AB treatment also significantly reduced CRC cells survival and dramatically sensitized cancer cells to etoposide-induced cell death, as evidenced by reduced cell viability and clonogenicity, in LRP16 upregulated SW480 cells (Figure 2—figure supplement 3D). These data further suggest that LRP16 accounts for the limited or lack of response to the cytotoxic and cytostatic effects of etoposide and protect cells from the death induced by genotoxic agents in a PAR-dependent manner. These results have been included in the revised manuscript.

3) The biological function of LRP16 mutants (D160A or I161A) on CRC cells should be performed and then further confirm NF-κB activation mediated by LRP16 plays crucial roles in tumor cells resistance to DNA damage therapies.

We appreciate the reviewer’s comments. We have conducted the experiments suggested by the reviewer. Our molecular mechanism studies on LRP16 mutants (D160A or I161A) showed that the function of LRP16 in genotoxic stresses-induced NF-κB activation dependent on its PAR binding ability. To further investigate the biological function of LRP16 mutants (D160A or I161A), we have performed cell viability and clonogenicity experiments in CRC cell lines. Of note, ectopic expression of LRP16 mutants (D160A or I161A), which were sufficient to significantly reduce its affinity to PAR, but not LRP16 wide-type (WT), in SW480 cells, dramatically sensitized cells to genotoxic stress-induced apoptosis, as conveyed by reduced cell viability and clonogenicity, in line with the indispensable role of the affinity of LRP16 to PAR in NF-κB activation and anti-apoptotic transcription (Figure 2—figure supplement 3D). These data further suggest that LRP16 plays a critical role in DNA damage-initiated and PAR-dependent NF-κB transactivation which could account for the limited or lack of response to the cytotoxic and cytostatic effects of etoposide and protect cells from the death induced by genotoxic agents. These results have been included in the revised manuscript.

4) Determination of LRP16 levels, p65 phosphorylation and XIAP in some representative colorectal carcinoma tissues and adjacent normal colon tissues by Western blots are needed to further support their conclusions. Alternatively, the specificity of the antibodies used may be confirmed.

We agree with the reviewers who pointed out that additional experiments should be performed to further validate the levels of LRP16, p65 phosphorylation and XIAP in some representative colorectal carcinoma tissues and adjacent normal colon tissues by Western blots. As the point suggested by reviewers, in the revised manuscript, we have analyzed the levels of LRP16, p65 and phosphorylation-p65 in some CRC samples and adjacent normal tissues by Western blots and/or RT-qPCR, which were frozen in liquid nitrogen after surgical removal and maintained at –80 °C until mRNA and protein extraction. Our additional experiments, as well as presented results demonstrate that the levels of LRP16 and p65 were highly elevated in primary CRC tissues as compared with their adjacent normal tissues as determined by RT-qPCR and Western blots. Similar to the level of LRP16 expression, the level of phosphorylation of p65 (phospho-p65) at Ser536 expression, which represents the activated form of NF-κB, is also significantly higher in CRC tissues (Figure 1—figure supplement 1B). We have also analyzed whether the correlation between LRP16 and NF-κB target genes expression (XIAP and *BCL2L1*) does exist in the TCGA cohorts of patients with CRC. In our IHC data, we did not observe significant correlation between LRP16 and XIAP expression in these CRC samples. However, analysis of TCGA CRC RNA-seq dataset revealed that *LRP16* expression was also not significantly correlated with that of *BCL2L1*, but was inversely correlated with that of XIAP(Figure 1—figure supplement 1D). We speculate that a different trend of the relevance of LRP16 and XIAP expression in our and TCGA cohorts of patients with CRC might be attributable to the dynamic expression patterns of NF-κB target genes, not following a standardized protocol, differing in the number of CRC samples analyzed and in the inclusion criteria of patients, which makes it difficult to compare results between groups. However, it may not be straightforward to compare different clinical specimens in different studies conducted at different labs. Alternatively, the specificity of the antibodies used in Western blots have been confirmed. These results have been included in the revised manuscript.

5) The authors should further analyze whether LRP16 might similarly regulate NF-κB pathway at the transcriptional level under the same condition (e.g. κb-Luc activity).

We appreciate that the reviewers have suggested several very insightful experiments to further address the interesting question on how LRP16 contributes to DNA damage-induced NF-κB activation. As the reviewer suggested, we have examined the impact of LRP16 on NF-κB transcriptional activation under DNA damage conditions by analysing κB-Luc activity. Similar to our immunoblot results, luciferase assay using a κB-Luc element demonstrated that NF-κB transcriptional activity is altered in response to modulation of LRP16 upon DNA damage stimulation. Ectopic of LRP16 remarkably enhanced the levels of NF-κB-dependent luciferase reporter gene activity in both SW480 and LoVo cells following either etoposide or IR treatment. Conversely, LRP16 deficiency introduced by two siRNAs in both HCT116 and SW620 cells considerably diminishes the NF-κB transcriptional activity in response to etoposide or IR. Of note, this process is also in a PAR-dependent manner, ectopic expression of LRP16 WT, but not its mutants (D160A or I161A) dramatically enhanced the NF-κB transcriptional activity in response to etoposide or IR. These results have been included in the revised manuscript (Figure 3—figure supplement 3).

6) The authors investigated the toxicity of MRS2578 or a combination of MRS2578 and etoposide in mice. However, these conclusions were only obtained from liver or kidney tissues in mice models. Whether other organs (e.g. heart, spleen, colon, etc.) display some toxicity should be confirmed.

The reviewers are asking an excellent question. We have conducted the suggested experiments to further investigate the organ toxicities of MRS2578 or a combination of MRS2578 and etoposide in mice and have analyzed the toxicity of MRS2578 or a combination of MRS2578 and etoposide in mice. Mice were treated with increasing doses (0, 10, and 20 mg/kg, twice weekly) of MRS2578 or a combination of MRS2578 and etoposide via intraperitoneal route for 28 days. Hematoxylin and eosin (H&E) staining of multiple organs, including brain, heart, lung, liver, kidney, spleen and intestine, showed no significant toxicities in the mice treated with MRS2578 or a combination of MRS2578 and etoposide. These results have been included in the revised manuscript (Figure 7—figure supplement 3).